# How Well Can Preference Optimization Generalize Under Noisy Feedback?

**Shawn Im**                                                                                    *shawnim@cs.wisc.edu*
*Department of Computer Sciences*
*University of Wisconsin-Madison*

**Sharon Li**                                                                                    *sharonli@cs.wisc.edu*
*Department of Computer Sciences*
*University of Wisconsin-Madison*

**Reviewed on OpenReview:** *https://openreview.net/forum?id=8f5gRWwzDx*

## Abstract

As large language models (LLMs) advance their capabilities, aligning these models with human preferences has become crucial. Preference optimization, which trains models to distinguish between preferred and non-preferred responses based on human feedback, has become a crucial component for aligning LLMs. However, most existing works assume noise-free feedback, which is unrealistic due to the inherent errors and inconsistencies in human judgments. This paper addresses the impact of noisy feedback on preference optimization, providing generalization guarantees under these conditions. In particular, we consider noise models that correspond to common real-world sources of noise, such as mislabeling and uncertainty. Unlike traditional analyses that assume convergence, our work focuses on finite-step preference optimization, offering new insights that are more aligned with practical LLM training. We describe how generalization decays with different types of noise across levels of noise rates based on the preference data distribution and number of samples. Our analysis for noisy preference learning applies to a broad family of preference optimization losses such as DPO, IPO, SLiC, etc. Empirical validation on contemporary LLMs confirms the practical relevance of our findings, offering valuable insights for developing AI systems that align with human preferences.

## 1 Introduction

Preference optimization, particularly through human-provided feedback, has emerged as a popular approach to ensuring that AI systems behave effectively and safely. A key recipe to achieve alignment is through the collection of binary preferences on generated outputs. In practice, human annotators are presented with two responses to the same question and provide binary judgments (*e.g.,* preferred, non-preferred) based on the quality of responses. Then, preference optimization algorithms such as those in Rafailov et al. (2023); Azar et al. (2023); Zhao et al. (2023); Tang et al. (2024) align the LLMs guided by the collected preference data. Preference-based alignment has demonstrated considerable success in enhancing the safety and usability of LLMs, making it a foundational component in the development of real-world LLM systems (OpenAI, 2023; Anthropic, 2023; Touvron et al., 2023; Gemini et al., 2023).

However, most existing works on preference optimization operate under the assumption of noise-free feedback (Rafailov et al., 2023; Azar et al., 2023; Zhao et al., 2023). This assumption, while simplifying the problem, does not hold in real-world scenarios where human feedback is inherently noisy. Factors such as human error and uncertain preference contribute to this noise, potentially leading to suboptimal or even harmful outcomes if not properly accounted for. The practical implications of noisy feedback are significant, as they directly impact the reliability and safety of AI systems deployed in critical applications. Therefore,

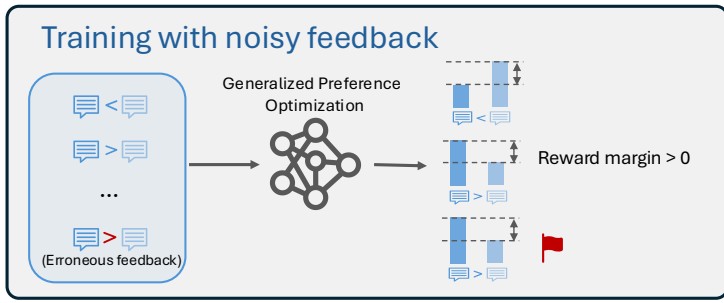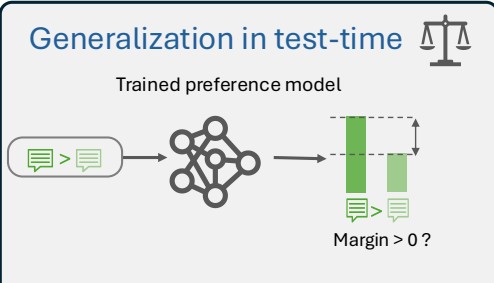

Figure 1: Overview of our work. The model is trained using the Generalized Preference Optimization loss with noisy human preference data (highlighted in red). Our framework provides theoretical guarantees on the generalization performance of the learned preference model.

understanding the effects of noisy feedback in preference optimization is crucial for the development of robust, aligned AI systems. Empirical evidence has shown that LLM performance can be highly sensitive to noise in preference labels and that the impact of noise can vary across datasets (Gao et al., 2024b), yet a rigorous theoretical understanding remains elusive. This raises a critical yet open research question:

> *How can we theoretically characterize the impact of noisy feedback on the generalization of preference optimization?*

In this work, we provide the first rigorous characterization of the generalization behavior of preference optimization under noisy human feedback and a practical training regime, filling a critical gap in the literature. Our analysis captures how generalization performance deteriorates with increasing noise, depending on both the structure of the data distribution and the training dynamics. We consider two realistic and representative noise models—the $\epsilon$-mislabeled model, which captures the effect of random label flips (e.g., due to human error), and the $\omega$-uncertain model, which captures uncertainty in human judgment by probabilistically sampling preferences based on an underlying reward margin. These models reflect the common sources of noise encountered in practice, such as annotator mistakes or ambiguous pairwise comparisons, and are supported by empirical observations in widely used datasets (Gao et al., 2024b). Our framework thus allows us to provide more realistic and practical guarantees for the generalization of preference optimization under realistic noise behavior, making our results relevant for the deployment of robust LLM systems.

In particular, we provide generalization guarantees for a broad family of preference optimization methods under noisy samples, encompassing existing algorithms such as DPO (Rafailov et al., 2023), IPO (Azar et al., 2023) and SLiC (Zhao et al., 2023) as special cases. All of these losses can be cast as a general form, referred to as generalized preference optimization (GPO) in Tang et al. (2024). Our guarantee captures how the generalization bound for GPO changes with the noise rate $\epsilon$, and based upon our theoretical results, we provide conditions under which model generalization is robust to noise and describe how the robustness behavior changes with dataset size and distributional properties. The key insight of our **Theorem** 3.3 is that for data that is well separated and with sufficient samples, as the noise rate increases, the population risk of the model remains near zero, demonstrating the value of dataset size and well distinguishable preferences. However, for data that is less separated or under high noise rates, the sample complexity significantly increases to maintain low error, demonstrating the challenges of preference learning for complex or inherently noisy data without noise-aware optimization. We empirically verify our theoretical insights on real-world Anthropic datasets (Perez et al., 2022), where we observe differences in performance decay based on distributional properties, demonstrating the practical relevance of our results. Overall, the close match between our theoretical analysis and empirical observation highlights the strength and applicability of our theoretical framework in modeling the effects of noise on preference optimization. Our contributions can be summarized as follows:

1. We establish the first generalization guarantees for preference optimization under noisy feedback. Our guarantees can be broadly applicable to *a generalized family of preference optimization ap-*

*proaches* (Tang et al., 2024), including DPO (Rafailov et al., 2023), IPO (Azar et al., 2023), SLiC (Zhao et al., 2023) as special cases.

2. We provide a comprehensive theoretical analysis of the impact of noise rate in the finite-step learning setting, leading to general and practically relevant insights that describe the effect of noise on generalization and the effect of the data distribution on sample efficiency for robustness guarantees across various settings.

3. We conduct comprehensive empirical evaluations that support our theoretical findings, showcasing the practical implications of our work.

## 2 Preliminaries on Preference Optimization

We denote $\pi_\theta$ as a language model policy parameterized by $\theta$, which takes in an input prompt $x$, and outputs a discrete probability distribution $\pi_\theta(\cdot|x)$ over the vocabulary space $\mathcal{V}$. $\pi_\theta(y|x)$ refers to the model's probability of outputting response $y$ given input prompt $x$. Preference optimization typically operates on pairs of responses and learns to discern the preferred response. Formally, we define the preference data below.

**Definition 2.1** (**Preference data**). *Consider two responses $y_w, y_l$ for an input prompt $x$, we denote $y_w \succ y_l$ if $y_w$ is preferred over $y_l$. We call $y_w$ the preferred response and $y_l$ the non-preferred response. Each triplet $(x, y_w, y_l)$ is referred to as a preference. Furthermore, the empirical dataset $\mathcal{D} = \{(x_i, y_{w,i}, y_{l,i})\}_{i=1}^N$ consists of $N$ such triplets sampled from a preference distribution.*

**Generalized Preference Optimization (GPO).** Recent work by Tang et al. (2024) presented a unified view of preference optimization encompassing existing algorithms, including DPO (Rafailov et al., 2023), IPO (Azar et al., 2023) and SLiC (Zhao et al., 2023) as special cases. All of these losses can be cast as a general form, referred to as generalized preference optimization (GPO):

$$\mathbb{E}_{(x,y_w,y_l)\in\mathcal{D}}\left[f\left(\beta\left(\log\frac{\pi_\theta(y_w|x)}{\pi_{\text{ref}}(y_w|x)} - \log\frac{\pi_\theta(y_l|x)}{\pi_{\text{ref}}(y_l|x)}\right)\right)\right], \tag{1}$$

where $\beta$ is a scalar and the function $f : \mathbb{R} \to \mathbb{R}$ can be instantiated differently:

- **DPO**: $f(z) = -\log\sigma(z)$ applies the logistic loss (Hastie et al., 2009).

- **IPO**: $f(z) = (z - 1/2)^2$ applies the squared loss (Azar et al., 2023).

- **SLiC**: $f(z) = \max(0, 1 - z)$ applies the hinge loss function (Zhao et al., 2023).

The above objective can also be connected to the implicit reward model given by $r(x,y) = \beta\log\frac{\pi_\theta(y|x)}{\pi_{\text{ref}}(y|x)}$. The implicit reward indicates how much more likely a given response is to be generated by the trained model $\pi_\theta$ compared to the base model $\pi_{\text{ref}}$. The implicit reward should be greater for preferred responses and smaller for non-preferred responses.

In this paper, our theoretical analysis revolves around this **generalized formulation**, and thus can be broadly applicable to preference optimization losses in the GPO family. We consider a set of objectives where $f(z)$ is a function with (i) $f'(0) < 0$ and $|f''(z)|$ bounded for all $x \geq 0$ or (ii) $f$ is the Hinge Loss as in SLiC. We define $D$ as $\sup_{x\geq 0}|f''(z)|$ if $f$ satisfies (i) and we can set $D = \frac{1}{2\beta}$ for (ii).

**Generalization of GPO.** From Equation (1), we can see that GPO learns to have a positive ***reward margin*** for a given sample $(x, y_w, y_l)$:

$$r_\theta(x, y_w, y_l) = \underbrace{\beta\left(\log\frac{\pi_\theta(y_w|x)}{\pi_\theta(y_l|x)} - \log\frac{\pi_{\text{ref}}(y_w|x)}{\pi_{\text{ref}}(y_l|x)}\right)}_{\text{Reward Margin}} > 0. \tag{2}$$

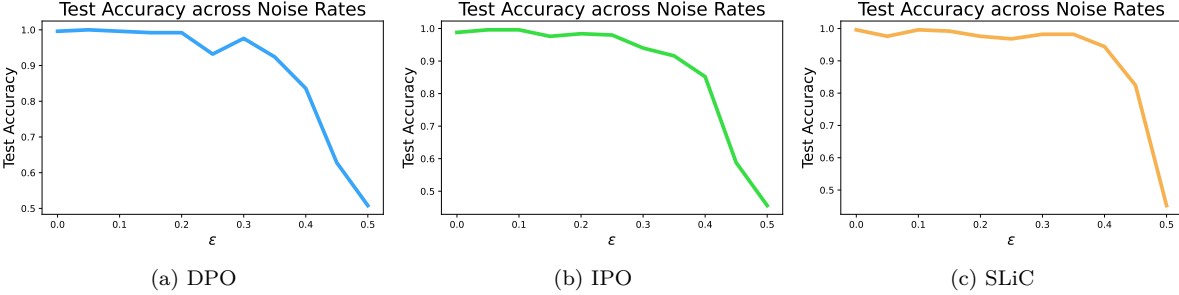

Figure 2: Test accuracy when trained on Anthropic dataset (behavior "willingness to make acausal trades") with varying noise levels $\epsilon$. We train Llama-3.1-8B using DPO, IPO, and SLiC and average the accuracy over 10 runs.

Under the notion of reward margin, the population risk can also be defined formally below based on the notion of the reward margin.

**Definition 2.2** (**Population risk of preference learning**). *We define the population risk in terms of a 0-1 loss where a sample's loss is 0 when the reward margin is positive and 1 otherwise.*

$$\mathcal{R}(x, y_w, y_l) = \left\{ \begin{array}{ll} 0 & r_\theta(x, y_w, y_l) > 0 \\ 1 & r_\theta(x, y_w, y_l) \leq 0 \end{array} \right.$$

*where $r_\theta(x, y_w, y_l)$ is the reward margin for a new sample $(x, y_w, y_l)$. Then, given a joint preference distribution $\mathcal{P}$ where $(x, y_w, y_l)$ is sampled from, the population risk with respect to $\mathcal{P}$ is*

$$\mathcal{R}(\mathcal{P}) = \mathbb{E}_{(x, y_w, y_l) \sim \mathcal{P}} \left[ \mathcal{R}(x, y_w, y_l) \right]. \tag{3}$$

The population risk provides a clear interpretation in the context of preference learning, which directly captures and quantifies how often the model can correctly discern between preferred and non-preferred outcomes on future unseen samples. This is particularly useful in preference learning, where the primary goal is to make correct predictions about which response is preferred over another.

## 3 How Well Can GPO Generalize Under Noisy Feedback?

### 3.1 Models of Noisy Feedback for Preference Optimization

Human feedback data is rarely flawless. In the real world, annotators can introduce noise for various reasons—ranging from labeling mistakes to uncertainty in judgment. For example, recent works have raised broader concern that the human feedback labeling in popular datasets such as HH-RLHF contains more than 25% of noise (Yeh et al., 2025; Wang et al., 2024). Such noisy feedback can significantly influence the preference optimization process, leading to suboptimal performance. In Figure 2, we illustrate empirically how noisy feedback impacts the generalization of preference learning objectives. To theoretically characterize and understand this, we begin by providing formal definitions for noisy preference datasets. For completeness, we consider a family of noise models—including both $\epsilon$-mislabeled and $\omega$-uncertain preference datasets—that capture realistic scenarios in preference learning, such as label flipping due to annotator errors or ambiguous comparisons.

**Definition 3.1** ($\epsilon$**-mislabeled preference data**). *An $\epsilon$-mislabeled preference dataset $\tilde{\mathcal{D}}_\epsilon = \{(x_i, \tilde{y}_{w,i}, \tilde{y}_{l,i})\}_{i=1}^N$ flips the preference label with probability $\epsilon$, from $y_w \succ y_l$ to $y_l \succ y_w$ for samples in the noise-free dataset $\mathcal{D} = \{(x_i, y_{w,i}, y_{l,i})\}_{i=1}^N$. $\epsilon$ captures the amount of noise in the dataset, where a larger $\epsilon$ means more severe noise contamination, and vice versa.*

**Definition 3.2** ($\omega$**-uncertain preference data**). *An $\omega$-uncertain preference dataset $\tilde{\mathcal{D}}_\omega = \{(x_i, \tilde{y}_{w,i}, \tilde{y}_{l,i})\}_{i=1}^N$ samples the preference label $y_{w,i} \succ y_{l,i}$ with probability $\sigma((r^*(y_{w,i}|x_i) - r^*(y_{l,i}|x_i))/\omega)$*

and $y_{l,i} \succ y_{w,i}$ *otherwise where* $r^*(\cdot)$ *is the true reward model.* $\omega$ *captures the level of uncertainty or ambiguity in the dataset, where a larger* $\omega$ *means preference labels are closer to a random guess, and vice versa.*

Given a noisy preference dataset $\tilde{\mathcal{D}}_\epsilon$ or $\tilde{\mathcal{D}}_\omega$, where $\tilde{\mathcal{D}}_{\epsilon/\omega} = \{(x_i, \tilde{y}_{w,i}, \tilde{y}_{l,i})\}_{i=1}^N$ denotes either $\epsilon$-mislabeled or $\omega$-uncertain data, we fine-tune the LLM policy $\pi_\theta$ by minimizing the GPO objective:

$$\mathbb{E}_{(x,\tilde{y}_w,\tilde{y}_l) \in \tilde{\mathcal{D}}_{\epsilon/\omega}} \left[ f\left( \beta \left( \log \frac{\pi_\theta(\tilde{y}_w|x)}{\pi_{\text{ref}}(\tilde{y}_w|x)} - \log \frac{\pi_\theta(\tilde{y}_l|x)}{\pi_{\text{ref}}(\tilde{y}_l|x)} \right) \right) \right], \tag{4}$$

where $\tilde{y}_w$ and $\tilde{y}_l$ are the noisy preferred and rejected labels for preference learning.

## 3.2 Analyzing Generalization via Boundary Dynamics

**Connection to practices.** A key focus of our paper is to provide a tractable analysis of GPO's generalization behavior under practical considerations. Our analytical framework is designed with practicality in mind. Besides taking realistically noisy feedback into account, we consider the generalization of models after finite gradient steps when the loss is within a constant factor of its initial value, extending the approach in Im & Li (2025) to general objective functions and noisy feedback. This scenario closely matches real-world practices, where large language models are often fine-tuned for a finite number of steps to avoid overfitting. For this reason, our analytical approach is different from classical generalization theory, which typically considers overparameterized models that achieve near-optimal loss (Allen-Zhu et al., 2019; Arora et al., 2019; Cao & Gu, 2020; Subramanian et al., 2022) or are independent of the training process (Arora et al., 2018; Lotfi et al., 2022; 2023).

Our theory revolves around analyzing how the decision boundary changes initially due to noise and over the course of training, which allows us to bound the generalization error after finite-step GPO updates. For an input prompt $x = (x^{(1)}, x^{(2)}, \ldots, x^{(T)})$ with length $T$, the model output is defined to be $\text{softmax}(W g(x^{(1)}, x^{(2)}, \ldots, x^{(T)}))$, where $g : \mathcal{V}^T \mapsto \mathbb{R}^d$ is the non-linear mapping from the prompt to the last hidden state, and $W \in \mathbb{R}^{|\mathcal{V}| \times d}$ is the unembedding layer matrix. This decomposition directly enables understanding how the latent preference distribution directly impacts the model's preference learning and its generalization properties, as we will present formally in Section 3.3. The feature backbone $g(x)$ can be either fixed or tunable, which will be comprehensively considered in our theory and experimental verification.

## 3.3 Generalization Guarantee

**Preference data distribution.** We now characterize the preference distribution in order to provide a tractable analysis and bound the generalization error. Importantly, the features we model are designed to reflect the characteristics of the real-world transformer backbone, ensuring that our theoretical analysis remains grounded in the specific inductive biases and structures that are typical of such models. Specifically, we find as seen in Table 1 that the embeddings after the RMSNorm layer in practical models such as Llama (Zhang & Sennrich, 2019; Touvron et al., 2023; Grattafiori et al., 2024) have near-uniform norm, suggesting that the embedding distribution can be closely modeled by a hyperspherical distribution. In particular, we consider the von Mises-Fisher (vMF) distribution, a classical and important distribution in directional statistics (Mardia & Jupp, 2009), which is analogous to spherical Gaussian distributions for features with unit norms (see Figure 3). The density function of the vMF distribution is given by $\rho(x; \mu, \kappa) = C_d(\kappa) e^{\kappa \mu^\top x}$, where $\mu$ represents the mean direction

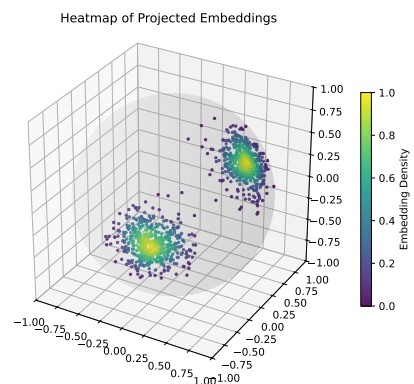

Figure 3: Visualization of vMF distributions corresponding to positive and negative examples on the 3-D sphere.

and $\kappa$ is the concentration parameter, and $C_d(\kappa)$ normalization constant dependent on the dimension $d$ and $\kappa$. We denote the distribution with mean direction $\mu$ and concentration parameter $\kappa$ as $\text{vMF}(\mu, \kappa)$. We also define a normalized concentration parameter $\gamma = \frac{2\kappa}{d}$.

**Empirical verification of assumption.** In Table 1, we verify that the embeddings from real-world models and datasets exhibit key characteristics of the vMF distribution. All embeddings are collected after RMSNorm has been applied, using the Llama-3.1-8B (Grattafiori et al., 2024) model on the Anthropic persona datasets (Perez et al., 2022). We collect the norm of the final layer embedding at the end of each statement and calculate both the average norm and the variance of the norm across all samples. We can see that *the norm is nearly uniform across samples* with a standard deviation smaller than 1% of the average, confirming the validity of our data assumption. Additionally, for every persona, we compute the mean embedding of the positive and negative samples and calculate the cosine similarity between each sample and its corresponding mean. We then average the cosine similarity for the positive samples and the negative samples, and compile these averages across all personas. The high average cosine similarity across datasets suggests that embeddings are indeed concentrated around a single direction.

Under this characterization, we can now describe the data-generating process. First, we generate the set of positive samples $\mathcal{D}_+$, consisting of $N/2$ *i.i.d.* samples from $\text{vMF}(\mu_+, \kappa)$ and the set of negative samples $\mathcal{D}_-$, consisting of $N/2$ *i.i.d.* samples from $\text{vMF}(\mu_-, \kappa)$. We define $2\phi$ to be the angle between $\mu_+$ and $\mu_-$ and have embedding dimension $d \geq 3$. We consider a clean dataset where positive samples have a preferred response token $y_+$ and a rejected response token $y_-$, while negative samples have the opposite preferences.

Table 1: Verification of vMF distribution.

| | |
|---|---|
| Average norm | 139.6 |
| Norm Variance | 0.9635 |
| Average cosine | 0.9557 |
| Cosine Variance | 8.963e-5 |

For each sample in an $\epsilon$-mislabeled dataset, we generate an *i.i.d.* sample from a Bernoulli distribution with parameter $\epsilon$, flipping the sample's label if the outcome is 1. This results in our noisy dataset $\tilde{\mathcal{D}}_\epsilon = \tilde{\mathcal{D}}_+ \cup \tilde{\mathcal{D}}_-$. For an $\omega$-uncertain dataset, we define the true reward model through $p(y_+ \succ y_-) = \sigma(\kappa(\mu_+ - \mu_-)^\top x)$, which corresponds to the conditional probability that $x$ is sampled from $\text{vMF}(\mu_+, \kappa)$. Then, the $\omega$-uncertain reward model that preferences are sampled from are $p(y_+ \succ y_-) = \sigma(\frac{\kappa}{\omega}(\mu_+ - \mu_-)^\top x)$. We define a corresponding noise rate $\epsilon_\omega$ for fixed $\gamma$ given by $\sigma\left(-\frac{\kappa}{\omega}\left(t_0(1 - \cos 2\phi) - \sin 2\phi\sqrt{1 - t_0^2}\right)\right)$, where $t_0 = \frac{\sqrt{\nu^2 + \kappa^2} - \nu}{2\kappa}$ and $\nu = (d - 3)/2$.

Through concentration results on the von Mises-Fisher distribution and control over the boundary over the course of training, which we prove in **Appendix B**, we are able to describe how the generalization changes with noise rate $\epsilon$, number of samples $N$, and on distributional parameters $\gamma, \phi$. All following results are expressed in terms of $\epsilon$-mislabeled datasets, but the same results hold for $\epsilon_\omega$ with $O(1/d)$ adjustments given that $t_0 > \cos(\phi)$. Since real-world models such as Llama have $d = 4096$, these terms are insignificant. We provide the corresponding results and proofs for $\epsilon_\omega$ in **Appendix C**.

**Theorem 3.3.** *(**Generalization guarantee under noisy feedback**)(Informal version of Theorem B.4) Under the setup described above, suppose we have an $\epsilon$-mislabeled dataset with $N \geq 25$ samples and $d \geq 64$, $\phi$ sufficiently large and satisfying $0 \leq \tan \phi \leq \sqrt{\log N}$, and $\delta$ sufficiently small. Then, with probability at least $1 - \frac{4}{N} - \frac{8}{N^2 d^2}$, for $0 < t \leq \frac{\sin(\delta)\tau}{4\beta^2 D}$ where $\tau$ is an inverse learning rate and for*

$$0 \leq \epsilon \leq \frac{1}{2} - \mathcal{O}\left(\frac{2 + \gamma}{\gamma}\sqrt{\frac{\log N}{N}}\right), \tag{5}$$

*the population risk is bounded as*

$$\mathcal{R}(\mathcal{P}) \leq c \exp\left(-\frac{d\gamma^2}{5(2 + \gamma)}\right) \tag{6}$$

*for a constant $c > 0$. Additionally, for any $N, \gamma, \phi$, we have that the expected value over the sampled noisy datasets, $\tilde{\mathcal{D}}$, of the population risk of the model, which we denote by $\mathbb{E}_{\tilde{\mathcal{D}}_\epsilon}[\mathcal{R}(\mathcal{P})]$ satisfies*

$$\mathbb{E}_{\tilde{\mathcal{D}}_\epsilon}[\mathcal{R}(\mathcal{P})]\Big|_{\epsilon=1/2} = \frac{1}{2} \qquad \frac{d^2}{d\epsilon^2}\mathbb{E}_{\tilde{\mathcal{D}}_\epsilon}[\mathcal{R}(\mathcal{P})]\Big|_{\epsilon=1/2} = 0 \tag{7}$$

**Theoretical insight.** Unlike classical generalization theory, which focuses on model behavior at convergence, our framework uses a finite-step analysis to expose how noise affects learning dynamics during fine-tuning. This allows us to precisely characterize the role of data properties—especially concentration and sample size—in achieving robustness to label noise. We summarize the key takeaways below:

> **Key takeaways of Theorem**
>
> 1. When the data distribution is sufficiently concentrated and separated, then the population risk can be guaranteed to remain exponentially small and near zero—as long as the noise rate satisfies $\epsilon < \frac{1}{2} - \mathcal{O}(\frac{2+\gamma}{\gamma}\sqrt{\frac{\log N}{N}})$.
>
> 2. Stronger concentration $\gamma$ and contrasting directions $\phi$ for positive and negative samples allow for tighter bounds and slower degradation in accuracy as the noise rate increases with better sample efficiency. In low-concentration regimes or less-separated settings, the sample complexity for maintaining robustness up to noise rate $\epsilon$ significantly increases with $N$ growing at least as $\tilde{\mathcal{O}}(1/\gamma^2(1-2\epsilon)^2)$.
>
> 3. As the noise rate approaches $\epsilon = \frac{1}{2}$, the risk exhibits an approximate *linear* increase toward $\frac{1}{2}$, forming an inflection point.

**Remarks.** We discuss how we expect our theorem to generalize under different assumptions such as those considered in adversarial attack literature. One common setting is where instead of having i.i.d. noise, we have that at most $N^c$ out of $N$ samples are continimated where $0 < c < 1$. In this setting, we expect an increase in the risk bound compared to our current theorem with due to the need to consider the worst case arrangement. The increase in risk would be dependent on the variance or other geometrical constraints on the distribution. Another setting that may be considered is where each sample can be contaminated with probability $\epsilon_i$. In this setting, the current theorem would apply with $\epsilon$ set to the average sample noise rate since we control the number of contaminated samples using Hoeffding's inequality.

**Practical implications.** Theorem 3.3 provides practical guidance for deploying preference optimization in real-world systems where noisy human feedback is common. By quantifying how the generalization performance degrades with increasing noise and characterizing the threshold at which learning fails, our results offer practical guidance on the data quantity and quality required for successful preference optimization. In particular, our theoretical insight suggests a practical diagnostic: before applying a preference optimization algorithm, practitioners can first examine the geometric properties of their dataset. If the data exhibits low concentration or poor separation, the theory implies that standard optimization may be highly sensitive to noise, and a noise-aware or robust preference learning algorithm such as Dr. DPO (Wu et al., 2024) should be preferred. In contrast, if the data is well-structured, even noisy labels may still yield strong generalization with standard methods. Thus, our framework provides a principled way to inform algorithmic choices based on dataset properties. One example of a dataset property that can be easily computed is the difference in embeddings between positive and negative samples or preferred and non-preferred responses to estimate separability. We note that these properties are model-dependent and as a result, a dataset with well-defined preferences can exhibit poor separation in embedding space due to the model's representations. As a result, these properties should be analyzed at the beginning of training once the model has been selected.

## 4 Connecting Theory to Practice

To understand how our theory guides practical LLM training, we verify the generalization behavior of preference optimization when updating last-layer parameters and updating all model parameters. In particular, Section 4.1 focuses on experiments conducted within a controlled setting, which allows us to systematically verify the impact of the noise rate and distributional properties on model performance and robustness. In Section 4.2, we extend our investigation to real-world datasets to validate the practical applicability of our findings in a more complex and realistic setting.

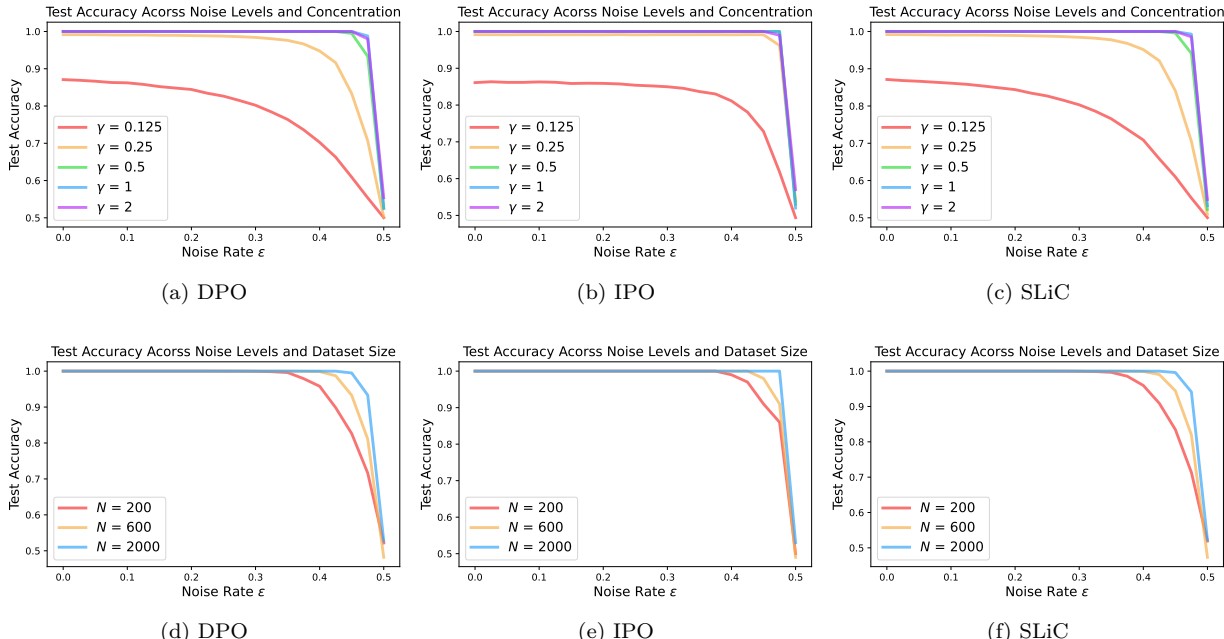

Figure 4: Empirical validation in a controlled setting using the DPO **(a), (d)**, IPO **(b), (e)**, and SLiC **(c), (f)** with (top) concentration parameter $\gamma$ varying over $1/8, 1/4, 1/2, 1, 2$ and with (bottom) $N$ varying over $200, 600, 2000$. We vary the noise rate $\epsilon$ on the $x$-axis from 0 to 0.5 with increments of 0.025. **All curves are averaged over 100 runs**.

## 4.1 Verification of Bound in a Controlled Setting

**Experimental setup.** We first validate the risk bound in a controlled setting where we can flexibly parameterize the data distribution. We consider data points with dimension $d = 512$, sampled from the vMF distribution, with the mean vectors for the positive and negative samples separated by an angle of $2\phi$. To study the effects of $\gamma$ and $N$, we vary the concentration parameter $\gamma$ over values $1/8, 1/4, 1/2, 1$, and $2$ while keeping $\phi$ fixed at $\pi/3$ for mislabeled datasets and at $\pi/2$ for uncertain datasets, ensuring $t_0 > \cos(\phi)$, and $N$ fixed at $2000$. We vary $N$ over $200, 600, 2000$ with $\gamma$ fixed at $1/2$. We sample $N/2$ data points each from the positive and negative distributions, with $\epsilon$ ranging from 0 to $1/2$ in increments of 0.025 for mislabeled datasets and with values of $\omega$ corresponding to the same set of $\epsilon$. We train a linear model with two outputs corresponding to positive and negative samples for 10 epochs using gradient descent with the DPO, IPO, and SLiC loss. For each configuration, we perform 100 trials and plot the average test accuracy on a test set of 2000 samples as a function of $\epsilon$. We present the results for mislabeled datasets in Figure 4 and the results for uncertain datasets in Appendix D.

**Impact of noise rate.** In Figure 4, we plot how the test accuracy of the model changes with increasing noise rate $\epsilon$. The figure aligns with our theoretical analysis of how the generalization error in preference learning increases as the noise rate rises. In particular, we can observe that the empirical accuracy observed decreases at a slower rate for smaller $\epsilon$ and is near zero for higher concentration values, validating the theoretical robustness guarantee. Additionally, we observe an inflection point around $\epsilon = 0.5$, where the test accuracy begins to decrease approximately linearly. These results align with the theoretical insights, further supporting that for moderate levels of noise (small $\epsilon$) and well separated data (large $\gamma$, large $\phi$), the model can still yield performance comparable to noiseless feedback, especially with more samples (large $N$), shedding light on why noisy feedback and synthetic data can remain useful for preference optimization. The results also indicate that for less separated data or data with inherently high noise rates, a significantly larger number of samples is required to achieve robustness, and preference learning remains challenging.

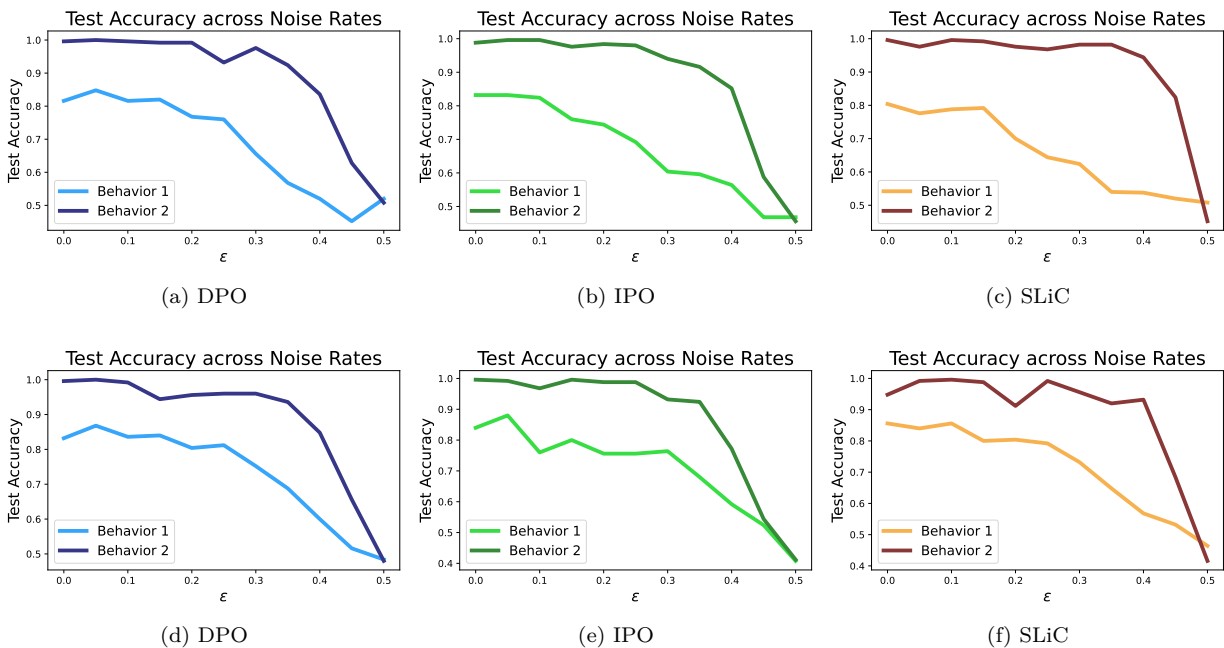

Figure 5: Empirical validation on the Anthropic datasets with varying preference separation using the DPO, IPO, and SLiC losses. Behavior 1 ("desire to remove safety precautions") is less separated, and Behavior 2 ("willingness to make acausal trades") is more separated. The top row corresponds to mislabeled datasets and the bottom row corresponds to uncertain datasets. **All curves are averaged over 10 runs**.

## 4.2 Verification on Real-World Dataset

**Experimental setup.** To further verify our theory on a real-world dataset, we consider the Anthropic Evaluations datasets (Perez et al., 2022). Unlike the HH-RLHF dataset, which already contains approximately 25-30% inherent noise (Wang et al., 2024), the Anthropic Evaluations datasets provide a cleaner baseline at $\epsilon = 0$, allowing us to systematically introduce and analyze the full trend of increasing noise levels. We plot the test accuracies of models trained on two behaviors with varying levels of separation across different noise rates and for different objective functions such as DPO, IPO, and SLiC. The level of separation is determined by the difference between the positive mean embedding and the negative mean embedding for each behavior. We train with noise rates ranging from $\epsilon = 0$ to $\epsilon = 0.5$ with 0.05 increments for mislabeled datasets and corresponding values of $\omega$ for uncertain datasets, and measure the test performance for each setting. We use 90% of the dataset for training and 10% for testing. We perform DPO/IPO/SLiC for 2 epochs on the noisy datasets. We provide the complete training hyperparameters in Appendix A and additional details on uncertain datasets in Appendix D.

**Our theoretical implication holds on real-world datasets with full fine-tuning.** We can see in Figure 5 that the empirical average test accuracy observes similar trends as in the controlled setting. Larger separation between the preference pairs allows for performance to remain robust up to a larger noise rate, while the behavior with smaller separation more quickly approaches a linear decline. All behaviors observe an inflection point at $\epsilon = 0.5$ where the more separated data have a higher slope.

This similarity in trends observed in real-world datasets and the expected behavior from our theorem further validates our theoretical framework. Our theoretical contribution offers a characterization of the behavior of test accuracy as $\epsilon$ increases, explaining when models can be robust to noise or when performance declines linearly and aligns with the empirical results. Overall, the close match between our theoretical analysis and empirical observation highlights the strength and applicability of our theoretical framework in modeling the effects of noise on preference optimization.

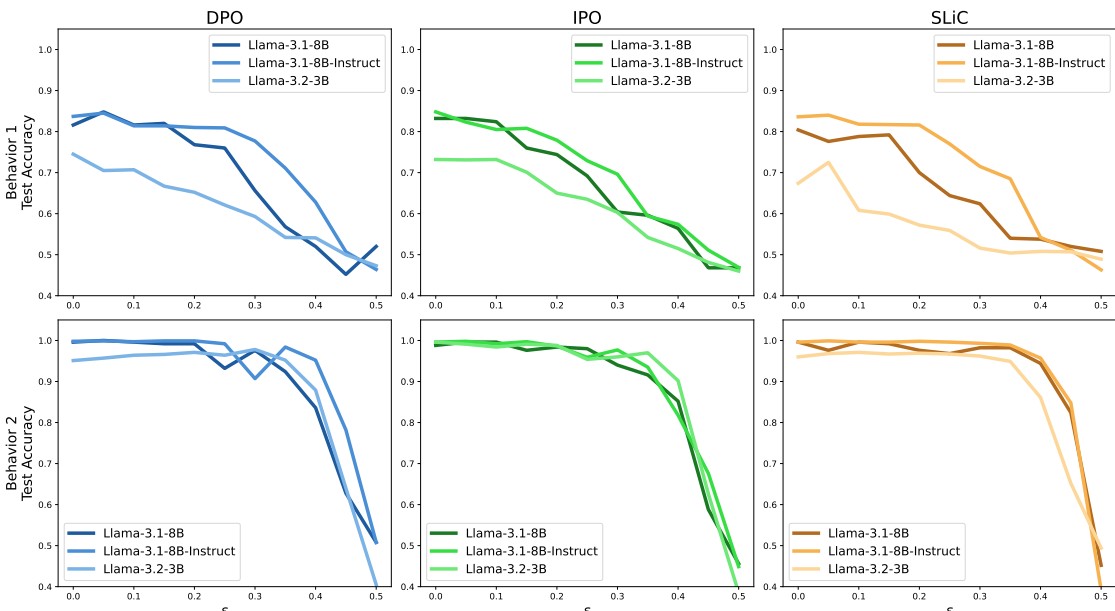

Figure 6: Comparison of robustness across models using the DPO, IPO, and SLiC losses. Behavior 1 ("desire to remove safety precautions") is less separated, and Behavior 2 ("willingness to make acausal trades") is more separated. The top row corresponds to Behavior 1 and the bottom row corresponds to Behavior 2 datasets. **All curves are averaged over 10 runs**.

**Effect of model representations.** We provide an analysis of how robustness to noise changes across models. Our theory and empirical results suggest that as the embeddings corresponding to a preference dataset become better separated, the model will be more robust to noise. The extent of separation depends not only on the dataset but also on the model. We compare the robustness of Llama-3.1-8B, Llama-3.1-8B-Instruct, and Llama-3.2-3B to mislabeling on the same personas as in Section 4.2 to see how behavior varies between a base model, an instruction-tuned model, and a smaller model. We provide the results in Figure 6, where we can see that the instruction-tuned model performs best while the smaller model performs the worst for both behaviors. We use the difference between the means of the last layer embeddings from the final prompt token of the positive and negative samples as a simple measure of separation, which we provide in Table 2. We can see that the instruction-tuned model has the largest separation while the smaller model has the smallest separation. We also can observe that beyond a certain level of separation, robustness shows marginal improvements. Based on these observations, we see that separability is model-dependent but consistent trends can be observed across different architectures.

## 5 Related Works

**Alignment of LLMs.** A key aspect of training and deploying large language models is ensuring the models behave in safe and helpful ways (Ji et al., 2023; Casper et al., 2023; Hendrycks et al., 2021; Leike et al., 2018). This is an important problem due to the potential harms that can arise in large models (Park et al., 2023; Carroll et al., 2023; Perez et al., 2022; Sharma et al., 2023; Bang et al., 2023; Hubinger et al., 2019; Berglund et al., 2023; Ngo et al., 2022; Shevlane et al., 2023; Shah et al., 2022; Pan et al., 2022). A wide range of methods have been developed that utilize human feedback or human preference data to train models to

Table 2: Verification of vMF distribution.

| Model | Separation (Behavior 1) |
|---|---|
| Llama-3.2-3B | 3.329 |
| Llama-3.1-8B | 5.090 |
| Llama-3.1-8B-Instruct | 7.209 |

| Model | Separation (Behavior 2) |
|---|---|
| Llama-3.2-3B | 10.308 |
| Llama-3.1-8B | 18.259 |
| Llama-3.1-8B-Instruct | 19.461 |

avoid harmful responses and elicit safer or more helpful responses (Christiano et al., 2017; Ziegler et al., 2019; Stiennon et al., 2020; Lee et al., 2021; Ouyang et al., 2022; Bai et al., 2022a; Nakano et al., 2022; Glaese et al., 2022; Snell et al., 2023; Yuan et al., 2023; Song et al., 2023; Dong et al., 2023; Bai et al., 2022b; Lee et al., 2023; Munos et al., 2023; Hejna et al., 2023; Dai et al., 2023; Khanov et al., 2024). Particularly, the Reinforcement Learning from Human Feedback (RLHF) framework has proven effective in aligning large pre-trained language models (Christiano et al., 2017; Ziegler et al., 2019; Ouyang et al., 2022; Bai et al., 2022a). However, given its computational inefficiency, recent shifts in focus favor closed-form losses that directly utilize offline preferences, like Direct Preference Optimization (DPO) (Rafailov et al., 2023) and related methodologies (Azar et al., 2023; Pal et al., 2024; Liu et al., 2024b; Ethayarajh et al., 2024a; Xiong et al., 2023; Tang et al., 2024; Meng et al., 2024; Ethayarajh et al., 2024b; Zeng et al., 2024; Calandriello et al., 2024; Muldrew et al., 2024; Ray Chowdhury et al., 2024; Liu et al., 2024a; Gao et al., 2024a; Yang et al., 2024; Chakraborty et al., 2024). Despite the empirical success and wide adoption in real-world systems (OpenAI, 2023; Anthropic, 2023; Touvron et al., 2023), fewer works provide theoretical underpinnings (Azar et al., 2023; Rafailov et al., 2024; Im & Li, 2024; Tang et al., 2024; Ray Chowdhury et al., 2024; Tajwar et al., 2024; Xu et al., 2024; Nika et al., 2024; Xiong et al., 2024; Im & Li, 2025) or consider the reward or comparison outcome as stochastic (Dumoulin et al., 2023; Siththaranjan et al., 2023). In this work, we make an initial attempt to theoretically analyze the generalization behavior of preference optimization under noisy feedback, making our results particularly relevant for the development and deployment of robust LLM systems.

**Robustness of preference optimization.** Ensuring that a model can generalize when trained with noisy labels is crucial for building robust and reliable systems (Song et al., 2022). This problem has led to a wide range of works (Song et al., 2022) developing various methods that improve model generalization in the presence of noise, with many of the works presenting theoretical guarantees of robustness (Natarajan et al., 2013; Zhang & Sabuncu, 2018; Li et al., 2020) for modified loss functions or for early stopping. Our study of pairwise preferences also has connections to binary classification with noisy data, where works such as Menon et al. (2018); Liu et al. (2022) focus on the impact of noise for risk minimizers. In the context of preference learning, increased noise levels have been shown to degrade performance, especially when considering loss minimizers (Gao et al., 2024b; Fisch et al., 2024; Liang et al., 2024), and model predictions can change significantly even when the probabilities of preferences in the training set change a small amount (Xu & Kankanhalli, 2025). This has led to the development of methods such as ROPO (Liang et al., 2024), cDPO (Mitchell, 2023), rDPO (Ray Chowdhury et al., 2024), and Dr. DPO (Wu et al., 2024), which introduce modifications to the DPO objective and its gradients. Fisch et al. (2024) considers a pessimistic distillation loss to learn rewards robustly. Kong et al. (2024) uses the perplexity difference between preferred and rejected responses to detect and correct noisy preferences. These approaches have proven effective in enhancing the robustness of preference optimization. Very recently, Yeh & Li (2025) created a benchmark called PrefCleanBench for data cleaning strategies to improve the reliability of preference-based LLM alignment under noisy feedback. Complementing these efforts, our study provides a rigorous generalization analysis of finite-step preference optimization under noisy feedback. Our theory, grounded in reward dynamics, offers new insights on how the population risk grows with the noise rate for offline preference learning in a finite-step training setting.

# 6 Conclusion

Our work theoretically analyzes the generalization behavior of preference learning in the presence of noisy labels through a dynamics-based approach based on a general class of objectives, including methods such as DPO, IPO, SLiC, etc., which implicitly learn a reward model. Key to our framework, we analyze the decision boundary and its trajectory throughout the training process, enabling us to effectively bound the generalization error. Through rigorous analysis and novel bounds, we establish a generalization guarantee that depends on the noise rate and distributional properties and provide practical insights based on the theoretical guarantee on sample that closely describe how test accuracy and sample efficiency for robustness are impacted by noise and dataset properties in real-world tasks using contemporary LLMs. By grounding our theory in practical regimes, our results lay the important foundation for provably robust preference optimization and catalyze future investigations into the theoretical understanding of preference optimization methods.

**Limitations and future work.** Despite its contributions to our understanding of noisy preference optimization, this work also has limitations that point to promising future research directions. One key constraint is our reliance on offline preference optimization methods, leaving open questions about how our insights might extend to online reinforcement learning settings, where feedback is integrated dynamically. Another promising direction is to extend our work to explore the generalization of noise-robust DPO objectives and compare their performance to that of common objectives within the GPO family. This enhanced understanding would enable practitioners to more confidently deploy preference optimization methods in real-world settings, where noisy feedback is the norm.

## Broader Impact Statement

Understanding how models learn from human preferences is an essential problem that addresses safety concerns around deploying machine learning models in the real world. Our research studies the effect of noise in human preference data and how this affects a model's ability to effectively generalize beyond the given preference data. Our findings provide practical insights for handling noisy preferences and when to apply noise-aware optimization.

## Acknowledgments

We thank Changdae Oh and Samuel Yeh for their valuable comments on the manuscript. This work is supported in part by the AFOSR Young Investigator Program under award number FA9550-23-1-0184, National Science Foundation under awards IIS-2237037 and IIS-2331669, Office of Naval Research under grant number N00014-23-1-2643, Schmidt Sciences Foundation, Open Philanthropy, Alfred P. Sloan Fellowship, and gifts from Google and Amazon. Shawn Im is also supported by the National Science Foundation Graduate Research Fellowship Program under Grant No. 2137424. Any opinions, findings, and conclusions or recommendations expressed in this material are those of the author(s) and do not necessarily reflect the views of the National Science Foundation. Support was also provided by the Graduate School and the Office of the Vice Chancellor for Research at the University of Wisconsin-Madison with funding from the Wisconsin Alumni Research Foundation.

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

## A  Additional Experimental Details

We provide the hyperparameters used for experiments.

Table 3: Summary of training hyperparameters for GPO on Llama-3.1-8B for the Anthropic Evaluation datasets.

| | Parameters | Value |
|---|---|---|
| | Number of epochs | 2 |
| | Optimizer | AdamW |
| | Learning rate | $10^{-5.5}$ |
| | $\beta$ | 0.1 |
| | Batch size | 100 |
| GPO | Gradient accumulation steps | 1 |
| | Maximum sequence length | 256 |
| | DeepSpeed Zero stage | 3 |
| | Max prompt length | 250 |
| | Max target length | 6 |
| | Weight decay | 0 |

**Software and hardware.**  We train with 4 A100 80GB GPUs using the TRL library (von Werra et al., 2020) and Huggingface library (Wolf et al., 2020) for full fine-tuning.

## B  Proof of Theorem 3.3

**Lemma B.1** (Gradient flow and reward dynamics)**.** *The dynamics of the reward margin for sample $j$ is given by*

$$\tau \dot{r_j}(t) = -\frac{1}{N} \sum_{i=1}^{N} \beta^2 \big( f'(r_i(t))(\tilde{\mathbf{y}}_{w,j} - \tilde{\mathbf{y}}_{l,j})^\top (\tilde{\mathbf{y}}_{w,i} - \tilde{\mathbf{y}}_{l,i}) \Sigma_{ij} \big), \tag{8}$$

*where $t$ is the time, $r_i$ is the shorthand notation for reward margin of sample $x_i$, $\Sigma$ is the sample covariance matrix with $\Sigma_{ij} = g(x_i)^\top g(x_j)$, and $\tau$ is inverse to the learning rate.*

**Proof.**  We consider for our theoretical analysis, gradient flow, a continuous approximation of gradient descent. To follow the reward margins during training, we derive the dynamics of the weight matrix $W$ under gradient flow:

$$\tau \dot{W} = \frac{1}{N} \sum_{i=1}^{N} \beta \sigma(-\beta(\mathbf{y}_{w,i} - \mathbf{y}_{l,i})^\top (W - W_0) g(x_i))(\mathbf{y}_{w,i} - \mathbf{y}_{l,i}) g(x_i)^\top, \tag{9}$$

where $\tau$ determines the rate of change, where a larger $\tau$ corresponds to a slower rate of change. Let $\Delta W = W - W_0$, a constant offset from $W$, we have:

$$\tau \Delta \dot{W} = \sum_{i=1}^{N} \beta \sigma(-\underbrace{\beta(\mathbf{y}_{w,i} - \mathbf{y}_{l,i})^\top \Delta W g(x_i)}_{\text{Reward margin for } x_i})(\mathbf{y}_{w,i} - \mathbf{y}_{l,i}) g(x_i)^\top, \tag{10}$$

which contains the term of the reward margin. Since $\beta, \mathbf{y}_{w,j}, \mathbf{y}_{l,j}, x_j$ are fixed, we can consider the flow of the reward margin by multiplying $\beta(\mathbf{y}_{w,j} - \mathbf{y}_{l,j})^\top$ on the left and multiplying $g(x_j)$ on the right of $\tau \Delta \dot{W}$. This yields the dynamics for the reward margin:

$$\tau \dot{r_j} = \frac{1}{N} \sum_{i=1}^{N} \beta^2 \sigma(-r_i)(\mathbf{y}_{w,j} - \mathbf{y}_{l,j})^\top (\mathbf{y}_{w,i} - \mathbf{y}_{l,i}) \Sigma_{ij}, \tag{11}$$

where $r_i$ is the shorthand notation for reward margin of sample $x_i$, and $\Sigma$ is the sample covariance matrix with $\Sigma_{ij} = g(x_i)^\top g(x_j)$.

**Lemma B.2** (Mislabeled von Mises-Fisher Directional Concentration). *Given an $\epsilon$-mislabeled dataset with $N \geq 25$ samples, with $\phi$ such that $0 \leq \tan(\phi) \leq \sqrt{\log N}$, and $0 \leq \epsilon \leq \frac{1}{2} - \frac{3(2+\gamma)}{2\gamma}\sqrt{\frac{\log N}{N}}$, we have with probability at least $1 - \frac{4}{N} - \frac{8}{N^2 d^2}$, the difference between the means of the positively labeled samples and the negatively labeled samples in the noisy dataset has cosine similarity with $\mu_+ - \mu_-$ of at least,*

$$1 - \frac{5}{\sqrt{N}} - \frac{5\sqrt{\frac{\log N}{N}}}{(1-2\epsilon)(\frac{\gamma}{2+\gamma}) + 2\sqrt{\frac{\log N}{N}}} \tag{12}$$

**Proof.** We will consider the difference of means between positively labeled samples and negatively labeled samples. We will do so by refactoring this difference as a weighted sum between the mean of noisy positive samples and noisy negative samples.

First, we consider the distribution of noisy positive samples where each sample can be written as $z_i x_i$ where $x_i$ is sampled from the positive vMF distribution and $z_i$ is a random variable with $1 - \epsilon$ probability of being 1 and $\epsilon$ probability of being $-1$. We start by considering the tangential component and lower bounding $\frac{2}{N}\sum_{i=1}^{N/2} z_i x_i^\top \mu_+$. Since by Laforgia & Natalini (2010), Theorem 1.1, $x_i^\top \mu_+$ has mean lower bounded by $t_\gamma = \frac{\sqrt{1+\gamma^2}-1}{\gamma}$, and is bounded between $-1$ and 1, we have by Hoeffding's inequality that with probability at least $1 - \frac{2}{N}$

$$\frac{2}{N}\sum_{i=1}^{N/2} z_i x_i^\top \mu_+ \geq (1-2\epsilon)\frac{\sqrt{1+\gamma^2}-1}{\gamma} - \sqrt{\frac{4\log N}{N}} \tag{13}$$

Now, we consider the tangential component of $\frac{2}{N}\sum_{i=1}^{N/2} z_i x_i^\top$ which is orthogonal to $\mu_+$. Since the tangential component for each $x_i$ is distributed uniformly over a $(d-2)$-dimensional hypersphere orthogonal to $\mu_+$, we have that norm of the average of the tangential component is upper bounded by $\frac{4}{N}$ with probability at least $1 - \frac{4}{N^2 d^2}$. We can ignore the sign from $z_i$ due to the symmetry in the uniform distribution. Then, the cosine of the angle between $\frac{2}{N}\sum_{i=1}^{N/2} z_i x_i$ and $\mu_+$ is lower bounded by

$$\frac{(1-2\epsilon)\frac{\sqrt{1+\gamma^2}-1}{\gamma} - \sqrt{\frac{4\log N}{N}}}{\sqrt{\left((1-2\epsilon)\frac{\sqrt{1+\gamma^2}-1}{\gamma} - \sqrt{\frac{4\log N}{N}}\right)^2 + \frac{16}{N^2}}} \tag{14}$$

and letting $t_{\epsilon,\gamma} = (1-2\epsilon)\frac{\sqrt{1+\gamma^2}-1}{\gamma} - \sqrt{\frac{4\log N}{N}}$ and furthering lower bounding the expression gives

$$\frac{t_{\epsilon,\gamma}}{t_{\epsilon,\gamma} + \frac{8}{N^2 t_{\epsilon,\gamma}}} \tag{15}$$

and further by

$$1 - \frac{8}{N^2 t_{\epsilon,\gamma}^2} \tag{16}$$

Since $\epsilon \leq \frac{1}{2} - \frac{3(2+\gamma)}{2\gamma}\sqrt{\frac{\log N}{N}}$, we have that this is further lower bounded by

$$1 - \frac{8}{N \log N} \tag{17}$$

Then, this occurs with probability at least $1 - \frac{2}{N} - \frac{4}{N^2 d^2}$.

Following the same argument as above, we have that the cosine of the angle between $\frac{2}{N}\sum_{i=1}^{N/2} z_i x_i$ and $\mu_-$ where now $z_i, x_i$ correspond to the noisy negative distribution, is lower bounded by lower bounded by

$$1 - \frac{8}{N\log N} \tag{18}$$

with probability at least $1 - \frac{2}{N} - \frac{4}{N^2 d^2}$. We will now lower bound the cosine similarity between the difference of means for noisy samples and $\mu_+ - \mu_-$. Let $a$ be the average of the noisy positive samples and $b$ be the average of the noisy negative samples. Let $a, b$ be within $\phi$ of $\mu_+, \mu_-$ respectively. Without loss of generality assume $\|b\| < \|a\|$ and let $r = \frac{\|b\|}{\|a\|}$. Furthermore, assume without loss of generality that $\mu_+ = e_1$ and $\mu_- = \cos 2\phi e_1 + \sin 2\phi e_2$ where $e_1, e_2$ are the first and second standard basis vectors respectively. Then, we consider the worst case which is when

$$a = \|a\|\cos\phi e_1 + \|a\|\sin\phi e_2 \tag{19}$$

and

$$b = r\|a\|\cos(2\phi + \phi)e_1 + r\|a\|\sin(2\phi + \phi)e_2 \tag{20}$$

Then, we have that

$$\frac{\langle a - b, \mu_+ - \mu_-\rangle}{\|a - b\|\,\|\mu_+ - \mu_-\|} = \frac{(1+r)(\cos\phi - \cos(2\phi - \phi))}{\sqrt{1 + r^2 - 2r\cos(2\phi)}\sqrt{2 - 2\cos(2\phi)}} \tag{21}$$

Using the half-angle formula and the sum of angles formulas, we have that

$$\frac{\langle a - b, \mu_+ - \mu_-\rangle}{\|a - b\|\,\|\mu_+ - \mu_-\|} = \frac{(1+r)(\cos\phi\sin\phi - \sin\phi\cos\phi)}{\sqrt{1 + r^2 - 2r\cos(2\phi)}} \tag{22}$$

In the worst case, using Hoeffding's inequality and the bound on the tangential component, letting $l_\mu(\gamma)$ be the expected radial component, we have that $\|a\| = (1 - 2\epsilon)l_\mu(\gamma) + 2\sqrt{\frac{\log N}{N}} + \frac{4}{N}$ and $\|b\| = (1 - 2\epsilon)l_\mu(\gamma) - 2\sqrt{\frac{\log N}{N}}$ so $r \geq 1 - \frac{4\sqrt{\frac{\log N}{N}} + \frac{4}{N}}{(1-2\epsilon)l_\mu(\gamma) + 2\sqrt{\frac{\log N}{N}} + \frac{4}{N}}$. Let $t_{\epsilon,\gamma} = \frac{4\sqrt{\frac{\log N}{N}} + \frac{4}{N}}{(1-2\epsilon)l_\mu(\gamma) + 2\sqrt{\frac{\log N}{N}} + \frac{4}{N}}$, then we have that

$$\frac{\langle a - b, \mu_+ - \mu_-\rangle}{\|a - b\|\,\|\mu_+ - \mu_-\|} \geq \frac{(2 - t_{\epsilon,\gamma})(\cos\phi\sin\phi - \sin\phi\cos\phi)}{\sqrt{2(1 - t_{\epsilon,\gamma})(1 - \cos(2\phi)) + t_{\epsilon,\gamma}^2}} \tag{23}$$

This is further lower bounded by

$$\frac{(1 - \frac{t_{\epsilon,\gamma}}{2})(\cos\phi - \sin\phi\tan\phi)}{\sqrt{1 - t_{\epsilon,\gamma}} + t_{\epsilon,\gamma}} \tag{24}$$

which is further lower bounded by

$$1 - \frac{4\sqrt{\frac{\log N}{N}} + \frac{4}{N}}{(1 - 2\epsilon)l_\mu(\gamma) + 2\sqrt{\frac{\log N}{N}} + \frac{4}{N}} - \frac{1 + 4\tan\phi}{\sqrt{N\log N}} \tag{25}$$

and by the restrictions on $\phi$,

$$1 - \frac{5}{\sqrt{N}} - \frac{4\sqrt{\frac{\log N}{N}} + \frac{4}{N}}{(1 - 2\epsilon)l_\mu(\gamma) + 2\sqrt{\frac{\log N}{N}} + \frac{4}{N}} \tag{26}$$

Lower bounding the result further using that $l_\mu(\gamma) \geq \frac{\gamma}{2+\gamma}$ and that $N \geq 25$, we have

$$1 - \frac{5}{\sqrt{N}} - \frac{5\sqrt{\frac{\log N}{N}}}{(1 - 2\epsilon)(\frac{\gamma}{2+\gamma}) + 2\sqrt{\frac{\log N}{N}}} \tag{27}$$

$\square$

**Lemma B.3** (Training Boundary Shift). *For $0 < t \leq \frac{\delta\tau}{4\beta^2 D}$, the angle between the boundary at time $t$ and the initial boundary is at most $\arcsin\delta$ for $0 < \delta < 1$.*

**Proof.** We start with the case for $f$ with $f'(0) < 0$ and $|f''(x)| \leq D$ for $x \geq 0$. As the weights follow the following dynamics,

$$\tau \Delta \dot{W} = -\frac{1}{N} \sum_{i=1}^{N} \beta f'(\underbrace{\beta(\tilde{\mathbf{y}}_{w,i} - \tilde{\mathbf{y}}_{l,i})^\top \Delta W g(x_i)}_{\text{Reward margin for } x_i})(\tilde{\mathbf{y}}_{w,i} - \tilde{\mathbf{y}}_{l,i})g(x_i)^\top, \tag{28}$$

we can say that the initial direction that the weights are along is

$$-\frac{1}{N} \sum_{i=1}^{N} \beta f'(0)(\tilde{\mathbf{y}}_{w,i} - \tilde{\mathbf{y}}_{l,i})g(x_i)^\top \tag{29}$$

which we will define as $W_{0+}$. We aim to control the angle between the initial boundary and the boundary at time $t$. To do so, consider any sample $x^*$ with corresponding reward $r^*$. Then, we know that at $t = 0$,

$$\tau \dot{r}^*(0) = \beta(\mathbf{y}_w^* - \mathbf{y}_l^*)^\top W_{0+} g(x^*). \tag{30}$$

Now, let $B_0 = (\mathbf{y}_w^* - \mathbf{y}_l^*)^\top W_{0+}$, and suppose the cosine similarity between $B_0, g(x^*)$ is greater than or equal to $\delta$. Then,

$$\tau \dot{r}^*(0) \geq \beta \|B_0\| \delta \tag{31}$$

Now, we will determine a lower bound, $t_s$, for $t^*$ which is defined as the first time $|\tau \dot{r}^*(t) - \tau \dot{r}^*(0)| = \beta \|B_0\| \delta$, and the lower bound should hold for any sample that satisfies the equation above as this will guarantee that the boundary shifts by at most an angle of $\arcsin \delta$ at time $t_s$. First, we bound the magnitude of the second time derivative of the reward which has the form

$$\tau \ddot{r}^*(t) = -\frac{1}{N} \sum_{i=1}^{N} \beta^2 f''(r_i)\dot{r}^*(t)(\mathbf{y}_w^* - \mathbf{y}_l^*)^\top (\tilde{\mathbf{y}}_{w,i} - \tilde{\mathbf{y}}_{l,i})g(x^*)^\top g(x_i) \tag{32}$$

Since we consider $f$ with second derivative with magnitude bounded by $D$ and unit norm embeddings,

$$|\ddot{r}^*(t)| \leq \frac{2\beta^2 D}{\tau}|\dot{r}^*(t)| \tag{33}$$

Since we consider time up to $t_s$, we know that $|\dot{r}^*(t)| \leq 2\beta \|B_0\|$. Then, it follows that

$$|\ddot{r}^*(t)| \leq \frac{4\beta^3 D \|B_0\|}{\tau^2} \tag{34}$$

Then, we have that

$$|\dot{r}^*(t) - \dot{r}^*(0)| \leq \frac{4\beta^3 D \|B_0\| t}{\tau^2} \tag{35}$$

Then, as we need $|\tau \dot{r}^*(t) - \tau \dot{r}^*(0)| \leq \beta \|B_0\| \delta$, we can lower bound $t_s$ by

$$\frac{\delta \tau}{4\beta^2 D} \tag{36}$$

Then, it follows that for $0 < t \leq \frac{\delta \tau}{4\beta^2 D}$, the angle between the boundary at time $t$ and the initial boundary is at most $\arcsin \delta$.

In the case of SLiC, since $f'(x) = 1$ for $0 \leq x < 1$, we can ensure that the boundary actually stays the same as initialization as long as we stop before any reward is greater than or equal to 1. We can ensure this by bounding $|\dot{r}^*(t)|$ for any sample $r^*$. Based on the fact that $f'(x) = 1$ for $0 \leq x < 1$ and that we will only have rewards in this range, we have that

$$|\dot{r}^*(t)| \leq \frac{2\beta}{\tau} \tag{37}$$

Then, since $\delta < 1$, at any time $0 < t \leq \frac{\delta \tau}{2\beta}$, $r^*(t) \leq \delta$ for any $r^*$, and since $\delta < 1$, we have that the boundary will not shift from the initial direction during this range of time. Then, since we set $D = \frac{1}{2\beta}$ for SLiC, this completes the proof. $\qquad \square$

**Theorem B.4.** *(**Generalization guarantee under noisy feedback**) Under the setup described, suppose we have an $\epsilon$-mislabeled dataset with $N \geq 25$ samples and that $d \geq 64$, and $\phi, \delta$ satisfy $0 \leq \tan \phi \leq \sqrt{\log N}$ and $\frac{\gamma}{5(\gamma+2)} - \cos \phi - 2\sqrt{\delta} > 0$. Then, with probability at least $1 - \frac{4}{N} - \frac{8}{N^2 d^2}$, for $0 < t \leq \frac{\sin(\delta)\tau}{4\beta^2 D}$ where $\tau$ is an inverse learning rate and for*

$$0 \leq \epsilon \leq \frac{1}{2} - \left( \frac{16}{(\frac{\gamma}{5(\gamma+2)} - \cos \phi - 2\sqrt{\delta})^2} + \frac{3}{2} \right) \frac{2+\gamma}{\gamma} \sqrt{\frac{\log N}{N}} \tag{38}$$

*the population risk is bounded as*

$$\mathcal{R}(\mathcal{P}) \leq c \exp\left( -\frac{d\gamma^2}{5(2+\gamma)} \right) \tag{39}$$

*for some constant $c > 0$. Additionally, for any $N, \gamma, \phi$, we have that the expected value over the sampled noisy datasets, $\tilde{\mathcal{D}}$, of the population risk of the model, which we denote by $\mathbb{E}_{\tilde{\mathcal{D}}_\epsilon}[\mathcal{R}(\mathcal{P})]$ satisfies*

$$\mathbb{E}_{\tilde{\mathcal{D}}_\epsilon}[\mathcal{R}(\mathcal{P})]\bigg|_{\epsilon=1/2} = \frac{1}{2} \qquad \frac{d^2}{d\epsilon^2} \mathbb{E}_{\tilde{\mathcal{D}}_\epsilon}[\mathcal{R}(\mathcal{P})]\bigg|_{\epsilon=1/2} = 0 \tag{40}$$

**Proof.** We start by upper bounding the probability that $\mu_+^\top x$ is less than or equal to some value $z < t_0$ where $t_0 = \frac{\sqrt{\gamma^2 + (1-\frac{3}{d})^2} - (1-\frac{3}{d})}{\gamma}$. By considering the radial distribution, we know that this is equal to

$$\frac{\int_{-1}^z e^{\kappa t + \frac{d-3}{2} \log(1-t^2)} dt}{\int_{-1}^1 e^{\kappa t + \frac{d-3}{2} \log(1-t^2)} dt} \tag{41}$$

For the denominator, we can apply Laplace's method as if we let the exponentiated term be $dg(t)$, we will see that $g(t)$ has a unique maximum and $g''(t) < 0$. Then, the result will have relative error $O(1/d)$. Doing so, we have an upper bound of

$$\frac{\int_{-1}^z e^{\kappa t + \frac{d-3}{2} \log(1-t^2)} dt}{e^{\kappa t_0 + \frac{d-3}{2} \log(1-t_0^2)} \sqrt{\frac{2\pi(1-t_0^2)^2}{(d-3)(1+t_0^2)}} (1 - O(1/d))} \tag{42}$$

Then, we can break the integral on top into two parts

$$\frac{\int_{-1}^0 e^{\kappa t + \frac{d-3}{2} \log(1-t^2)} dt + \int_0^z e^{\kappa t + \frac{d-3}{2} \log(1-t^2)} dt}{e^{\kappa t_0 + \frac{d-3}{2} \log(1-t_0^2)} \sqrt{\frac{2\pi(1-t_0^2)^2}{(d-3)(1+t_0^2)}} (1 - O(1/d))} \tag{43}$$

Since $\kappa t + \frac{d-3}{2} \log(1-t^2) < 0$ for $t \leq 0$, we have an upper bound of

$$\frac{1 + \int_0^z e^{\kappa t + \frac{d-3}{2} \log(1-t^2)} dt}{e^{\kappa t_0 + \frac{d-3}{2} \log(1-t_0^2)} \sqrt{\frac{2\pi(1-t_0^2)^2}{(d-3)(1+t_0^2)}} (1 - O(1/d))} \tag{44}$$

Then, using that from 0 to $t_0$, $\kappa t + \frac{d-3}{2} \log(1 - t^2)$ is positive and increasing, we can upper bound the numerator to get

$$\frac{1 + z e^{\kappa z + \frac{d-3}{2} \log(1-z^2)}}{e^{\kappa t_0 + \frac{d-3}{2} \log(1-t_0^2)} \sqrt{\frac{2\pi(1-t_0^2)^2}{(d-3)(1+t_0^2)}} (1 - O(1/d))} \tag{45}$$

We can upper bound the expression further by

$$\frac{2 e^{\kappa z + \frac{d-3}{2} \log(1-z^2)}}{e^{\kappa t_0 + \frac{d-3}{2} \log(1-t_0^2)} \sqrt{\frac{2\pi(1-t_0^2)^2}{(d-3)(1+t_0^2)}} (1 - O(1/d))} \tag{46}$$

and we can rewrite this as

$$\frac{2(1-t_0^2)\sqrt{2\pi}}{\sqrt{(d-3)(1+t_0^2)}} \exp\left(\kappa(z-t_0) + \frac{d-3}{2}\log\left(\frac{1-z^2}{1-t_0^2}\right)\right)(1+O(1/d)) \tag{47}$$

Then, given $d \geq 64$, this is upper bounded by

$$\exp\left(\kappa(z-t_0) + \frac{d-3}{2}\log\left(\frac{1-z^2}{1-t_0^2}\right)\right)(1+O(1/d)) \tag{48}$$

and further by

$$\exp\left(\frac{d}{2}\left(\gamma(z-t_0) + \log\left(\frac{1-z^2}{1-t_0^2}\right)\right)\right)(1+O(1/d)) \tag{49}$$

Then, we are interested in a lower bound for $z_*$ which satisfies

$$\gamma z_* + \log(1-z_*^2) = \gamma t_0 + \log(1-t_0^2) - \frac{2\log(1/B)}{d} \tag{50}$$

where $B = \frac{d\gamma^2}{5(\gamma+2)}$. Since $\log(1-z_*^2) < 0$, solving

$$\gamma z = \gamma t_0 + \log(1-t_0^2) - \frac{2B}{d} \tag{51}$$

provides a lower bound $z$ for $z_*$. Then, we have

$$z = t_0 + \frac{\log(1-t_0^2)}{\gamma} - \frac{2B}{d\gamma} \tag{52}$$

Then, using that $t_0 \geq \frac{\gamma}{2+\gamma}$ is a lower bound for $t_0$, and that $\log(1-t_0^2)/\gamma \geq -e^{-(1-3/d)}\gamma/(2+\gamma)$, and $d \geq 64$, we have that

$$z = \frac{3\gamma}{5(\gamma+2)} - \frac{2B}{d\gamma} \tag{53}$$

is also a lower bound for $z_*$. Then, for all $z \leq \frac{3\gamma}{5(\gamma+2)} - \frac{2B}{d\gamma}$, the probability of $\mu_+^\top x \leq z$ is at most $ce^{-B}$ for some constant $c > 0$ accounting for the $O(1/d)$ relative error. Now, we consider the worst case deviation in the boundary from $\mu_+ - \mu_-$. From Lemma B.2 and Lemma B.3, using that $\cos(x - \arccos(1-a)) \leq \cos(x) + 2\sin(x)\sqrt{a}$ and accounting for tangential deviations, we have that as long as

$$\mu_{+/-}^\top x_{+/-} \geq \cos\phi + 2\sqrt{\frac{5}{\sqrt{N}} + \frac{5\sqrt{\frac{\log N}{N}}}{(1-2\epsilon)(\frac{\gamma}{2+\gamma}) + 2\sqrt{\frac{\log N}{N}}}} + 2\sqrt{\delta} \tag{54}$$

the sample will classified correctly. Then, by considering an upper bound on the right-hand hand using $N \geq 25$, we have

$$\mu_{+/-}^\top x_{+/-} \geq \cos\phi + 4\sqrt{\frac{2}{(1-2\epsilon)(\frac{\gamma}{2+\gamma})}}\sqrt{\frac{\log N}{N}} + 2\sqrt{\delta} \tag{55}$$

Then, the expected risk will be less than or equal to $ce^{-B}$ if

$$\cos\phi + \left(\frac{4\sqrt{2}}{\sqrt{(1-2\epsilon)(\frac{\gamma}{2+\gamma})}}\left(\frac{\log N}{N}\right)^{1/4}\right) + 2\sqrt{\delta} \leq \frac{3\gamma}{5(\gamma+2)} - \frac{2B}{d\gamma} \tag{56}$$

Rearranging, we have

$$\frac{4\sqrt{2}}{\sqrt{(1-2\epsilon)(\frac{\gamma}{2+\gamma})}}\left(\frac{\log N}{N}\right)^{1/4} \leq \frac{3\gamma}{5(\gamma+2)} - \frac{2B}{d\gamma} - \cos\phi - 2\sqrt{\delta} \tag{57}$$

Reciprocating both sides and rearranging, we have

$$\sqrt{1-2\epsilon}\left(\frac{N}{\log N}\right)^{1/4} \geq \frac{4\sqrt{2}}{\left(\frac{3\gamma}{5(\gamma+2)} - \frac{2B}{d\gamma} - \cos\phi - 2\sqrt{\delta}\right)\sqrt{\frac{\gamma}{2+\gamma}}} \tag{58}$$

Squaring both sides and rearranging, we have

$$1-2\epsilon \geq \frac{32\sqrt{\log N}}{\left(\frac{\gamma}{2+\gamma}\right)\left(\frac{3\gamma}{5(\gamma+2)} - \frac{2B}{d\gamma} - \cos\phi - 2\sqrt{\delta}\right)^2\sqrt{N}} \tag{59}$$

and that for

$$\epsilon \leq \frac{1}{2} - \frac{16}{\left(\frac{\gamma}{2+\gamma}\right)\left(\frac{3\gamma}{5(\gamma+2)} - \frac{2B}{d\gamma} - \cos\phi - 2\sqrt{\delta}\right)^2}\sqrt{\frac{\log N}{N}} \tag{60}$$

the expected risk will be less than or equal to $ce^{-B}$. Then, using that $\epsilon \leq \frac{1}{2} - \frac{3(2+\gamma)}{2\gamma}\sqrt{\frac{\log N}{N}}$, we can have the same guarantee for

$$\epsilon \leq \frac{1}{2} - \left(\frac{16}{\left(\frac{3\gamma}{5(\gamma+2)} - \frac{2B}{d\gamma} - \cos\phi - 2\sqrt{\delta}\right)^2} + \frac{3}{2}\right)\frac{2+\gamma}{\gamma}\sqrt{\frac{\log N}{N}} \tag{61}$$

Now, we consider $\mathbb{E}_{\tilde{\mathcal{D}}_\epsilon}[\mathcal{R}(\mathcal{P})]$. Let $x_1, \ldots, x_N$ represent the sample embeddings and let $z_1, \ldots, z_N$ be $\text{Ber}(\epsilon)$ variables that determine the label flipping. Then,

$$\mathbb{E}_{\tilde{\mathcal{D}}_\epsilon}[\mathcal{R}(\mathcal{P})] = \int_{\mathbb{R}^d}\cdots\int_{\mathbb{R}^d}\mathbb{E}_{z_1,\ldots,z_N}[\mathcal{R}(\mathcal{P})|x_1,\ldots,x_N]\nu(x_1,\ldots,x_N)dx_1\ldots dx_N \tag{62}$$

where $\nu(x_1, \ldots, x_N)$ is the joint density of the sample embeddings. We can additionally expand $\mathbb{E}_{z_1,\ldots,z_N}[\mathcal{R}(\mathcal{P})|x_1,\ldots,x_N]$ as a sum over the $2^N$ possible $z_1, \ldots, z_N$ configurations. Since $\epsilon$ appears only within the sum and the sum is polynomial in $\epsilon$, we know that $\mathbb{E}_{\tilde{\mathcal{D}}_\epsilon}[\mathcal{R}(\mathcal{P})]$ is twice differentiable in $\epsilon$ as we can move $\frac{d^2}{d\epsilon^2}$ inside the integral and inside the sum. Now, we will show that

$$\mathbb{E}_{\tilde{\mathcal{D}}_\epsilon}[\mathcal{R}(\mathcal{P})]\big|_\epsilon = 1 - \mathbb{E}_{\tilde{\mathcal{D}}_\epsilon}[\mathcal{R}(\mathcal{P})]\big|_{1-\epsilon} \tag{63}$$

Since $\nu(x_1, \ldots, x_N)$ is independent of $\epsilon$, the above is true if for a given $x_1, \ldots, x_N$,

$$\mathbb{E}_{z_1,\ldots,z_N}[\mathcal{R}(\mathcal{P})|x_1,\ldots,x_N] = 1 - \mathbb{E}_{z'_1,\ldots,z'_N}[\mathcal{R}(\mathcal{P})|x_1,\ldots,x_N] \tag{64}$$

where $z_1, \ldots, z_N \sim \text{Ber}(\epsilon)$ and $z'_1, \ldots, z'_N \sim \text{Ber}(1-\epsilon)$. The probability of sampling a given $z_1, \ldots, z_N$ is the same as sampling $z'_1, \ldots, z'_N$ with the exact opposite set of labels being flipped. We know that the reward dynamics, for any sample $(x^*, y^*_w, y^*_l)$ and letting $r^*$ be its reward margin, follow

$$\tau\dot{r}^* = \frac{1}{N}\sum_{i=1}^N \beta^2 f'(r_i)(\mathbf{y}^*_w - \mathbf{y}^*_l)^\top(\tilde{\mathbf{y}}_{w,i} - \tilde{\mathbf{y}}_{l,i})g(x^*)^\top g(x_i) \tag{65}$$

Additionally, the reward dynamics for the training samples are the same for $z_1, \ldots, z_N$ and $z'_1, \ldots, z'_N$, so the reward dynamics for any new sample are the exact opposite for $z_1, \ldots, z_N$ and $z'_1, \ldots, z'_N$. This means that the resulting models have exact opposite predictions and therefore

$$\mathbb{E}_{z_1,\ldots,z_N}[\mathcal{R}(\mathcal{P})|x_1,\ldots,x_N] = 1 - \mathbb{E}_{z'_1,\ldots,z'_N}[\mathcal{R}(\mathcal{P})|x_1,\ldots,x_N] \tag{66}$$

Now, since we know that

$$\mathbb{E}_{\tilde{\mathcal{D}}_\epsilon}[\mathcal{R}(\mathcal{P})]\big|_\epsilon = 1 - \mathbb{E}_{\tilde{\mathcal{D}}_\epsilon}[\mathcal{R}(\mathcal{P})]\big|_{1-\epsilon} \tag{67}$$

we have that

$$\mathbb{E}_{\tilde{\mathcal{D}}_\epsilon}[\mathcal{R}(\mathcal{P})]\bigg|_{\epsilon=1/2} = \frac{1}{2} \tag{68}$$

and we can apply $\frac{d^2}{d\epsilon^2}$ to both sides and we have that

$$\frac{d^2}{d\epsilon^2}\mathbb{E}_{\tilde{\mathcal{D}}_\epsilon}[\mathcal{R}(\mathcal{P})]\bigg|_\epsilon = -\frac{d^2}{d\epsilon^2}\mathbb{E}_{\tilde{\mathcal{D}}_\epsilon}[\mathcal{R}(\mathcal{P})]\bigg|_{1-\epsilon} \tag{69}$$

and at $\epsilon = 1/2$, this is only possible if

$$\frac{d^2}{d\epsilon^2}\mathbb{E}_{\tilde{\mathcal{D}}_\epsilon}[\mathcal{R}(\mathcal{P})]\bigg|_{\epsilon=1/2} = 0 \tag{70}$$

$\square$

## C    Proofs under Uncertainty Model

**Lemma C.1.** *For $d > 3$, the expected fraction of samples with label $y_- \succ y_+$ for $x \sim vMF(\mu_+, \kappa)$ is upper bounded by*

$$\sigma\left(-\frac{\kappa}{\omega}\left(t_0(1 - \cos 2\phi) - \sin 2\phi\sqrt{1 - t_0^2}\right)\right) + O(1/d) \tag{71}$$

*Similarly, the expected fraction of samples with label $y_+ \succ y_-$ for $x \sim vMF(\mu_-, \kappa)$ is upper bounded by*

$$\sigma\left(-\frac{\kappa}{\omega}\left(t_0(1 - \cos 2\phi) - \sin 2\phi\sqrt{1 - t_0^2}\right)\right) + O(1/d) \tag{72}$$

**Proof.**    We start with the first case of $x \sim \text{vMF}(\mu_+, \kappa)$. We start by converting the expression to be in terms of $t = \mu_+^\top x$, so that we can use the radial component distribution. We know that the angle between $x$ and $\mu_+$ is $\arccos(t)$ and so $\mu_-^\top x$ is at most $\cos(2\phi - \arccos(t))$. Then, we have that

$$\sigma(\frac{\kappa}{\omega}(\mu_+ - \mu_-)^\top x) \geq \sigma(\frac{\kappa}{\omega}(t - \cos(2\phi - \arccos(t)))) \tag{73}$$

Applying the difference of angles formula, we have

$$\sigma(\frac{\kappa}{\omega}(\mu_+ - \mu_-)^\top x) \geq \sigma\left(\frac{\kappa}{\omega}(t(1 - \cos(2\phi)) + \sin(2\phi)\sqrt{1 - t^2})\right) \tag{74}$$

Now, we have that

$$\mathbb{E}_{x \sim \text{vMF}(\mu_+, \kappa)}[\sigma(\frac{\kappa}{\omega}(\mu_+ - \mu_-)^\top x)] \geq \mathbb{E}_t\left[\sigma\left(\frac{\kappa}{\omega}(t(1 - \cos(2\phi)) + \sin(2\phi)\sqrt{1 - t^2})\right)\right] \tag{75}$$

where $\mathbb{E}_t$ is the expectation over the radial component of the vMF distribution. We can write the right hand side as an integral

$$\frac{1}{C}\int_{-1}^{1}\sigma\left(\frac{\kappa}{\omega}(t(1 - \cos(2\phi)) + \sin(2\phi)\sqrt{1 - t^2})\right)e^{\kappa t}(1 - t^2)^{(d-3)/2}dt \tag{76}$$

where $C$ is a normalizing constant for the radial density function. Writing $\kappa = \frac{\gamma d}{2}$, we have

$$\frac{1}{C}\int_{-1}^{1}\sigma\left(\frac{\kappa}{\omega}(t(1 - \cos(2\phi)) + \sin(2\phi)\sqrt{1 - t^2})\right)\exp\left(\frac{\gamma dt}{2} + \frac{d-3}{2}\log(1 - t^2)\right)dt \tag{77}$$

We can see that the term being exponentiated is proportional to $d$ and if we can apply Laplace's method, it would result in an $O(1/d)$ relative error. We check the conditions for applying Laplace's method. Let $g(x) = \frac{\gamma t}{2} + \left(\frac{1}{2} - \frac{3}{2d}\right)\log(1 - t^2)$. We have that

$$g'(t) = \frac{\gamma}{2} - \left(1 - \frac{3}{d}\right)\frac{t}{1 - t^2} \tag{78}$$

$$g''(t) = -\left(1 - \frac{3}{d}\right)\frac{t^2 + 1}{(1 - t^2)^2} \tag{79}$$

We see that $g''(t) < 0$ for $d > 3$ and that $g'(t) = 0$ has a unique solution at

$$t_0 = \frac{\sqrt{\gamma^2 + \left(1 - \frac{3}{d}\right)^2} - \left(1 - \frac{3}{d}\right)}{\gamma} \tag{80}$$

Then, applying Laplace's method, we have that $\mathbb{E}_t\left[\sigma\left(\frac{\kappa}{\omega}(t(1 - \cos(2\phi)) + \sin(2\phi)\sqrt{1 - t^2})\right)\right]$ is

$$\frac{1}{C}\sigma\left(\frac{\kappa}{\omega}(t_0(1 - \cos(2\phi)) + \sin(2\phi)\sqrt{1 - t_0^2})\right)e^{\kappa t_0}(1 - t_0^2)^{(d-3)/2}(1 + O(1/d)) \tag{81}$$

Since, we can apply Laplace's method to the density function and also get an $O(1/d)$ relative error, we can factor out the exponential term and the normalizing constant together while maintaining the same level of precision. This gives

$$\mathbb{E}_{x \sim \text{vMF}(\mu_+, \kappa)}[\sigma(\frac{\kappa}{\omega}(\mu_+ - \mu_-)^\top x)] \geq \sigma\left(\frac{\kappa}{\omega}(t_0(1 - \cos(2\phi)) + \sin(2\phi)\sqrt{1 - t_0^2})\right)(1 + O(1/d)) \quad (82)$$

Then, the expected fraction of samples to flip is upper bounded by

$$1 - \sigma\left(\frac{\kappa}{\omega}(t_0(1 - \cos(2\phi)) + \sin(2\phi)\sqrt{1 - t_0^2})\right)(1 + O(1/d)) \quad (83)$$

or

$$\sigma\left(-\frac{\kappa}{\omega}(t_0(1 - \cos(2\phi)) + \sin(2\phi)\sqrt{1 - t_0^2})\right) + O(1/d) \quad (84)$$

The exact same argument can be applied for $\sigma(\frac{\kappa}{\omega}(\mu_- - \mu_+)^\top x)$ for $x \sim \text{vMF}(\mu_-, \kappa)$. $\qquad\square$

**Remark.** Due to the $O(1/d)$ term, when $\omega = 0$, we know that at most $O(1/d)$ samples will be past the angle bisector of $\mu_+, \mu_-$. We will follow the proof for the mislabeled datasets but with an additional $O(1/d)$ term adjustment to any part that relies on $1/2$ of the samples being initially positive or negative.

**Theorem C.2.** *(Generalization guarantee under noise from uncertainty) Under the setup described, suppose we have an $\omega$-mislabeled dataset with corresponding noise rate $\epsilon_\omega$ with $N \geq 25$ samples and that $d \geq 64$, and $\phi, \delta$ satisfy $0 \leq \tan\phi \leq \sqrt{\log N}$ and $\frac{\gamma}{5(\gamma+2)} - \cos\phi - 2\sqrt{\delta} > 0$. Then, with probability at least $1 - \frac{4}{N} - \frac{8}{N^2 d^2}$, for $0 < t \leq \frac{\sin(\delta)\tau}{4\beta^2 D}$ where $\tau$ is an inverse learning rate and for*

$$0 \leq \epsilon_\omega \leq \frac{1}{2} - \left(\frac{16}{(\frac{\gamma}{5(\gamma+2)} - \cos\phi - 2\sqrt{\delta})^2} + \frac{3}{2}\right)\frac{2+\gamma}{\gamma}\sqrt{\frac{\log N}{N}} - \frac{C}{d}, \quad (85)$$

*the population risk is bounded as*

$$\mathcal{R}(\mathcal{P}) \leq c\exp\left(-\frac{d\gamma^2}{5(2+\gamma)}\right) \quad (86)$$

*for some constants $C, c > 0$. Additionally, for any $N, \gamma, \phi$, we have that the expected value over the sampled noisy datasets, $\tilde{\mathcal{D}}$, of the population risk of the model, which we denote by $\mathbb{E}_{\tilde{\mathcal{D}}_\epsilon}[\mathcal{R}(\mathcal{P})]$ satisfies*

$$\mathbb{E}_{\tilde{\mathcal{D}}_\epsilon}[\mathcal{R}(\mathcal{P})]\Big|_{\epsilon_\omega = 1/2} = \frac{1}{2} \qquad \frac{d^2}{d\epsilon_\omega^2}\mathbb{E}_{\tilde{\mathcal{D}}_\epsilon}[\mathcal{R}(\mathcal{P})]\Big|_{\epsilon_\omega = 1/2} = 0 \quad (87)$$

**Proof.** Since the probability of the label corresponding to the distribution mean being positively correlated with the cosine similarity between the sample and the mean, and therefore the distribution of samples conditioned on their label being correct is more concentrated away from the boundary. Furthermore, the probability of the label is independent of tangential components outside the plane of $\mu_+, \mu_-$. We can additionally use Lemma B.3 directly as it is independent of the data distribution. Furthermore, as $\epsilon_\omega$ is an upper bound on the noise rate, we can apply the same arguments as with $\epsilon$-mislabeled datasets in Theorem 3.3.

Now, we consider $\mathbb{E}_{\tilde{\mathcal{D}}_\epsilon}[\mathcal{R}(\mathcal{P})]$. Let $x_1, \ldots, x_N$ represent the sample embeddings and let $z_1, \ldots, z_N$ be $\text{Ber}(\epsilon(x, \omega))$ variables that determine the label flipping where $\epsilon(x, \omega) = \sigma(-\frac{\kappa}{\omega}(\mu_+ - \mu_-)^\top x)$ if $x$ is from the positive distribution and $\epsilon(x, \omega) = \sigma(-\frac{\kappa}{\omega}(\mu_- - \mu_+)^\top x)$ otherwise. Then,

$$\mathbb{E}_{\tilde{\mathcal{D}}_\epsilon}[\mathcal{R}(\mathcal{P})] = \int_{\mathbb{R}^d} \cdots \int_{\mathbb{R}^d} \mathbb{E}_{z_1, \ldots, z_N}[\mathcal{R}(\mathcal{P})|x_1, \ldots, x_N]\nu(x_1, \ldots, x_N)dx_1 \ldots dx_N \quad (88)$$

where $\nu(x_1, \ldots, x_N)$ is the joint density of the sample embeddings. We can additionally expand $\mathbb{E}_{z_1, \ldots, z_N}[\mathcal{R}(\mathcal{P})|x_1, \ldots, x_N]$ as a sum over the $2^N$ possible $z_1, \ldots, z_N$ configurations. Since the sum consists of products of sigmoids with $1/\omega$ inside, after rewriting in terms of $\omega$, we know that $\mathbb{E}_{\tilde{\mathcal{D}}_\epsilon}[\mathcal{R}(\mathcal{P})]$ is twice differentiable in $\omega$, so we can move $\frac{d^2}{d\omega^2}$ inside the integral and inside the sum. Now, we will show that

$$\mathbb{E}_{\tilde{\mathcal{D}}_\epsilon}[\mathcal{R}(\mathcal{P})]\big|_\omega = 1 - \mathbb{E}_{\tilde{\mathcal{D}}_\epsilon}[\mathcal{R}(\mathcal{P})]\big|_{-\omega} \quad (89)$$

The above is true if for a given $x_1, \ldots, x_N$,

$$\mathbb{E}_{z_1,\ldots,z_N}[\mathcal{R}(\mathcal{P})|x_1,\ldots,x_N] = 1 - \mathbb{E}_{z'_1,\ldots,z'_N}[\mathcal{R}(\mathcal{P})|x_1,\ldots,x_N] \tag{90}$$

where $z_1, \ldots, z_N \sim \text{Ber}(\epsilon(x_i, \omega))$ and $z'_1, \ldots, z'_N \sim \text{Ber}(\epsilon(x_i, -\omega))$. The probability of sampling a given $z_1, \ldots, z_N$ is the same as sampling $z'_1, \ldots, z'_N$ with the exact opposite set of labels being flipped. We know that the reward dynamics, for any sample $(x^*, y_w^*, y_l^*)$ and letting $r^*$ be its reward margin, follow

$$\tau \dot{r^*} = \frac{1}{N} \sum_{i=1}^{N} \beta^2 f'(r_i)(\mathbf{y}_w^* - \mathbf{y}_l^*)^\top (\tilde{\mathbf{y}}_{w,i} - \tilde{\mathbf{y}}_{l,i}) g(x^*)^\top g(x_i) \tag{91}$$

Additionally, the reward dynamics for the training samples are the same for $z_1, \ldots, z_N$ and $z'_1, \ldots, z'_N$, so the reward dynamics for any new sample are the exact opposite for $z_1, \ldots, z_N$ and $z'_1, \ldots, z'_N$. This means that the resulting models have exact opposite predictions and therefore

$$\mathbb{E}_{z_1,\ldots,z_N}[\mathcal{R}(\mathcal{P})|x_1,\ldots,x_N] = 1 - \mathbb{E}_{z'_1,\ldots,z'_N}[\mathcal{R}(\mathcal{P})|x_1,\ldots,x_N] \tag{92}$$

Now, since we know that

$$\mathbb{E}_{\tilde{\mathcal{D}}_\epsilon}[\mathcal{R}(\mathcal{P})]\big|_\omega = 1 - \mathbb{E}_{\tilde{\mathcal{D}}_\epsilon}[\mathcal{R}(\mathcal{P})]\big|_{-\omega} \tag{93}$$

we have that

$$\mathbb{E}_{\tilde{\mathcal{D}}_\epsilon}[\mathcal{R}(\mathcal{P})]\bigg|_{\omega=\pm\infty} = \frac{1}{2} \tag{94}$$

and we can apply $\frac{d^2}{d\omega^2}$ to both sides and we have that

$$\frac{d^2}{d\omega^2}\mathbb{E}_{\tilde{\mathcal{D}}_\epsilon}[\mathcal{R}(\mathcal{P})]\bigg|_\omega = -\frac{d^2}{d\omega^2}\mathbb{E}_{\tilde{\mathcal{D}}_\epsilon}[\mathcal{R}(\mathcal{P})]\bigg|_{-\omega} \tag{95}$$

Then, changing variables to $\epsilon_\omega$ and taking the limits $\omega \to \pm\infty$, we must have

$$\frac{d^2}{d\epsilon_\omega^2}\mathbb{E}_{\tilde{\mathcal{D}}_\epsilon}[\mathcal{R}(\mathcal{P})]\bigg|_{\epsilon_\omega=1/2} = 0 \tag{96}$$

$\square$

# D Additional Results for Uncertain Datasets

## D.1 Controlled Setting Results

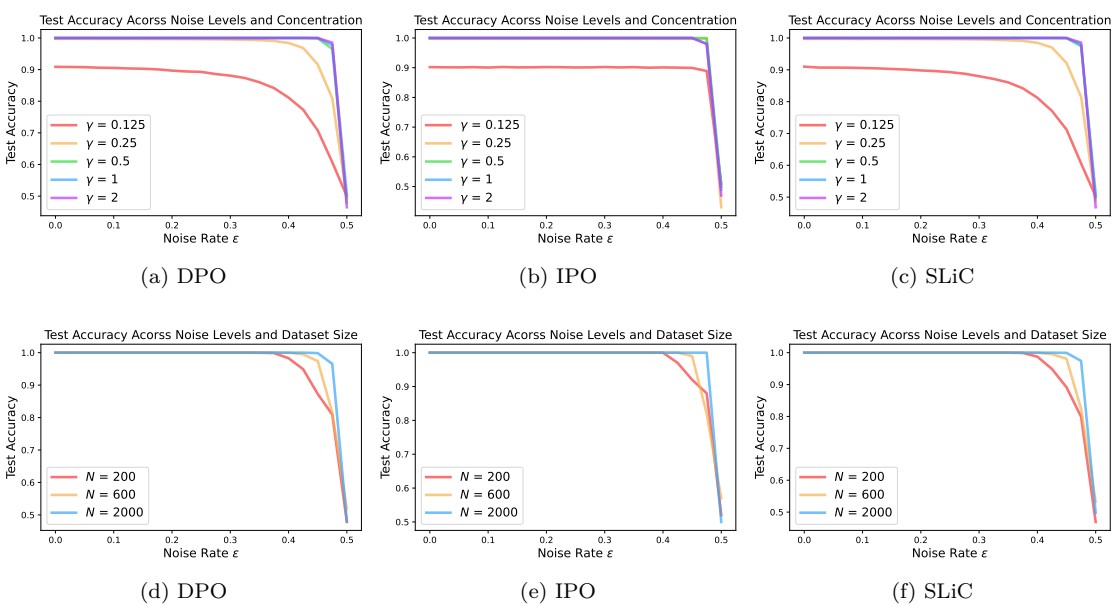

Figure 7: Empirical validation in a controlled setting using the DPO **(a), (d)**, IPO **(b), (e)**, and SLiC **(c), (f)** with (top) concentration parameter $\gamma$ varying over $1/8, 1/4, 1/2, 1, 2$ and with (bottom) $N$ varying over $200, 600, 2000$. We vary $\omega$ corresponding to the noise rate $\epsilon$ on the $x$-axis from 0 to 0.5 with increments of 0.025. All curves are averaged over 100 runs.

## D.2 Real-World Dataset Details

For each of the Anthropic Evaluation datasets, we train a linear classifier with binary cross-entropy loss on the final last-layer embedding of statements for the whole dataset to act as the true reward model. We train using Adam for 500 epochs with a learning rate of 0.01. We then perform a search over $\omega$ such that the average probability over the dataset of the label changing would be within 0.001 of $\epsilon$. For Behavior 1 (desire to remove safety precautions) the $\omega$ corresponding to $\epsilon$ from 0 to 0.5 with 0.05 increments are $0, 1.3125, 1.9921875, 2.75, 3.65625, 4.84375, 6.5, 9.125, 14.25, 29.0, \infty$, and for Behavior 2 (willingness to make acausal trades), $0, 7.9375, 11.75, 15.625, 20.25, 26.0, 34.25, 47.5, 73.0, 148.0, \infty$.

