# OpenReview forum: "How Well Can Preference Optimization Generalize Under Noisy Feedback?"
_TMLR — Accepted by TMLR_

### Review · Reviewer_DQoq · 2025-10-31

**Summary Of Contributions:**

The negative impact of noisy feedback on preference optimization is real. This paper investigates the issue in depth and provides various theoretical analyses. Overall, it is a well-balanced study that integrates both theory and practice. This area has already been studied by many researchers, as the authors note in the related work. However, since TMLR does not evaluate based on novelty alone, I believe this is a paper worth accepting due to its completeness and the interesting nature of its topic.

**Audience:**

Yes

**Audience Explanation:**

This is certainly a topic that will attract considerable interest.

**Claims And Evidence:**

Yes

**Claims Explanation:**

The paper is overall clear and logically structured.

**Requested Changes:**

The manuscript has no obvious weaknesses overall, but I have a few questions I would like to discuss with the authors.

1. Currently, most datasets seem to inject noise manually to validate the effectiveness of proposed models. I wonder if there are methods to quantitatively detect the proportion of noise in a dataset, or to identify noisy data points based on mechanistic interpretation. If a denoising method is evaluated solely based on whether it improves performance, it can be somewhat tricky.
2. Which existing denoising preference optimization methods are considered the best, and are there any simple methods that are generally effective in most cases?

---

> ### Author Response · Authors · 2025-11-22
>
> We thank the reviewer for your thoughtful comments and for recognizing the value of our work. We will provide responses to the questions provided below.
>
> 1. Methods to quantitatively detect noise
>
> In works such as [1], the noise rate of a dataset is estimated by dividing the dataset into subsets based on preference strength and **retraining a reward model on each subset**. Subsets that decrease accuracy are considered to be noisy. Other works, such as [2], have considered using **LLMs as a judge** to quantify the amount of noise or the consistency rate.
>
> On identifying noisy data points based on mechanistic interpretability, while works have considered using mechanistic interpretability to detect attacks, we are not aware of any works that aim to detect noise or inconsistencies in a dataset.
>
> 2. Denoising preference optimization
>
> For handling noise, there are methods such as rDPO [3] and ROPO [4], which modify DPO to handle noise but also introduce a noise-dependent hyperparameter. A more recent work, Dr.DPO [5], modifies the loss to be **robust to noise without an additional hyperparameter**, which suggests it may be effective to apply in most cases. While a comprehensive evaluation between methods has not been done, results suggest Dr.DPO can provide improved results over other methods. We have mentioned these methods in the updated **implications and related works sections**.
>
> [1] Wang, Binghai, et al. "Secrets of rlhf in large language models part ii: Reward modeling."
>
> [2] Wang, Zhaoyang, et al. "Cream: Consistency regularized self-rewarding language models."
>
> [3] Chowdhury, Sayak Ray, Anush Kini, and Nagarajan Natarajan. "Provably robust dpo: Aligning language models with noisy feedback."
>
> [4] Liang, Xize, et al. "ROPO: Robust Preference Optimization for Large Language Models."
>
> [5] Wu, Junkang, et al. "Towards robust alignment of language models: Distributionally robustifying direct preference optimization."

---

### Review · Reviewer_Xq8m · 2025-11-06

**Summary Of Contributions:**

This paper is the first to study preference optimization under noisy feedback (dataset contamination). It establishes a theoretical criterion that allows one to determine, given the amount of data and the level of contamination, whether successful preference optimization can be achieved with high probability. The authors further validate this criterion empirically using real-world datasets, demonstrating its practical effectiveness.

**Additional Comments:**

1. The assumption on data contamination (Definition 3.1) appears somewhat strong to me. Specifically, the paper assumes that each data point is contaminated independently with probability $\epsilon$. This effectively imposes two potentially restrictive conditions on the treated model: (i) the contamination events are independent across samples, and (ii) the contamination probabilities are identical, i.e., the data are assumed to be i.i.d. under contamination. In contrast, in some adversarial attack literature, a more flexible assumption is often adopted --- for instance, among $N$ samples, at most $N^c$ samples ($c \in (0,1)$) may be contaminated.
 Such an assumption relaxes the independence and identical-distribution requirements.  I am not suggesting that the authors need to revise their main results (Theorem 3.3) under this alternative setting. However, I would encourage them to include a discussion on how their analysis or guarantees might extend (or adapt) under this weaker contamination assumption, or to outline potential challenges for such an extension (if this extension is hard).

2. In addition, there is an even weaker assumption on the contamination model: the $i$-th data point ($i \in [N]$) is contaminated with probability $\epsilon_i$, where $\epsilon_i \in (0,1)$. Under this assumption, I am curious whether the proof of Theorem~3.3 can be adapted to yield a guarantee that depends on the average contamination level $\frac{1}{N}\sum_{i=1}^N \epsilon_i$. I do not require the authors to provide a full proof in this paper, but I would encourage them to discuss a possible proof strategy (if such an extension is possible), or to point out the main technical obstacles (if it is not).

3. The criterion proposed in this paper --- which determines the success of preference optimization based on the values of $N$ and $\epsilon$ --- assumes that $\epsilon$ (or a tight upper bound thereof) is known. In practice, however, this quantity may only be estimated heuristically or inferred from specific empirical conditions.

4. According to my understanding, increasing the size of the dataset can only enlarge the tolerance margin (i.e., allow for larger values of $\epsilon$ and for the desired event to hold with higher probability), but it would not reduce the population risk. I would like the authors to provide some proof-level insights on this aspect.

Overall, I appreciate both the theoretical and experimental contributions of this paper. I believe the authors do not necessarily need to provide new results; rather, they should offer more interpretation of the existing ones and discuss the potential extensions and challenges of their results.

**Audience:**

Yes

**Audience Explanation:**

This paper primarily investigates preference optimization, which is currently a highly active and relevant topic in the machine learning community. I believe that many members of the TMLR audience would find this work of significant interest.

**Claims And Evidence:**

Yes

**Claims Explanation:**

The paper provides rigorous theoretical analysis and supports its claims with empirical evidence on real-world datasets.

**Requested Changes:**

I would like the authors to include additional discussion, with specific suggestions provided in the Additional Comments section below.

---

> ### Author Response · Authors · 2025-11-22
>
> We thank the reviewer for your thoughtful comments and for recognizing the value of our work! We respectfully address the concerns below.
>
> 1. Assumption Discussion
>
> We appreciate the pointer to assumptions used in adversarial attack literature. Under the assumption that at most $N^c$ samples are contaminated, we would **expect similar but weaker results** than what would be obtained by setting $\epsilon = N^c$ since we now need to consider a worst-case arrangement. This would then result in an increase in the risk bound dependent on the variance or other geometrical constraints on the distribution. We have added this discussion to the updated manuscript on **page 7**.
>
> 2. Weaker Assumption
>
> In the case of having sample-wise contamination probabilities, the results should look the **same as the current result but with $\epsilon$ set to the average contamination level**. This is because one of the intermediate steps is bounding the number of samples flipped by Hoeffding's inequality, and we can still use the same technique. We have added this discussion to the updated manuscript on **page 7**.
>
> 3. Criterion
>
> While the criterion provided in the theorem does require knowing $\epsilon$, and in practice, this can only be estimated or inferred, we believe this is a useful starting point for informing decisions in practice. We provide an example of **our theorem can guide us towards developing simple metrics** such as the difference of means of the positive and negative examples to inform us about robustness to noise in **Section 4.2**
>
> 4. Dataset size vs tolerance margin and risk
>
> Our theorem presents a fixed risk bound with respect to the number of samples, with only the tolerance being dependent on $N$. The **fixed risk bound can be considered as a threshold for small population risk**, and the theorem bounds the noise range for which this small risk applies and where the risk vs. noise curve is flat. Beyond this range, **the population risk grows at a noticeably increasing rate up to the inflection point**.
>
> Using this threshold, we can reason about how properties such as sample size affect robustness. As an example, consider a noise rate $\epsilon'$ beyond the tolerance for small $N$ but less than the tolerance for large $N$. At $\epsilon'$, our theorem can tell us that
> 1. The population risk for large $N$ will be smaller than the risk for small $N$.
> 2. The risk for small $N$ is significantly increasing.
>
> However, in this range, providing a precise bound is difficult due to challenges in approximating the relevant integrals.

---

### Review · Reviewer_jRzx · 2025-11-10

**Summary Of Contributions:**

The paper studies how preference optimization methods (e.g., DPO, IPO, SLiC) behave when human preference data is noisy. It provides theoretical generalization analyses for a unified Generalized Preference Optimization (GPO) formulation. The obtained generalization bounds depend on dataset size, embedding concentration/separation, and noise level. Additionally, this work provides empirical validation on both synthetic vMF-distributed data and real-world Anthropic datasets, showing that
- When data is well-separated, performance remains stable even under moderate noise.
- When data is poorly separated, accuracy drops quickly as noise increases.

Overall, this paper argues that robustness to noisy preference data depends strongly on the structure and separability of the embedding space, not merely on dataset size.

**Audience:**

Yes

**Audience Explanation:**

The question of how preference optimization behaves under noisy or inconsistent human feedback is central to current LLM alignment practice and the interests of TMLR's audience.

**Broader Impact Concerns:**

The work has no direct negative societal impact in terms of generating harmful content or deploying unsafe systems. Instead, it contributes to the reliability and robustness of alignment techniques, which is broadly beneficial.

**Claims And Evidence:**

Yes

**Claims Explanation:**

Overall, this paper provides credible theoretical support and reasonable empirical alignment. Please refer to the requested changes for potential improvement.

**Requested Changes:**

1. It is nice to have the "Key takeaways of Theorem" and "Practical implications" discussions, but it would be more insightful and accessible if the implications were stated more operationally and executable. For example, be specific about how to "examine the geometric properties of their dataset" and examine whether "the data exhibits low concentration or poor separation", e.g., in a proposed algorithm.
2. In this paper, the notion of “separation” is model-dependent, not dataset-intrinsic. However, the model may not discover the true preference structure if two responses differ but the model collapses them. Therefore, I suggest adding the following clarifications:
   1. Make explicit in the main paper that separability is defined relative to the current model’s embedding space, not the dataset alone;
   2. Discuss the implications of cases where dataset-level preferences are well-defined, but the model’s representation space is not yet aligned with those distinctions;
   3. Provide guidance on whether separability should be evaluated before selecting a model, after selecting a model, or iteratively during training;
   4. To disentangle dataset-intrinsic signal from model-dependent representation quality, the authors could include the following experiment (Evaluate Separability Across Multiple Models):
      - Take the same preference datasets
      - Compute separation metrics using embeddings from:
         * The original pretrained model used in the paper (Llama-3.1-8B)
         * A smaller, less capable model
         * A larger or instruction-tuned model
         * An off-the-shelf sentence embedding model (e.g., BERT/SimCSE)
      - Run DPO/IPO/SLiC fine-tuning and compare noise robustness curves across models
      - Expected Insight: If the paper’s theory reflects dataset-intrinsic structure, separation rankings should be consistent across architectures. This experiment would identify whether the paper’s practical guideline is model-agnostic or model-specific, which is currently ambiguous.

---

> ### Author Response · Authors · 2025-11-22
>
> We thank the reviewer for their thoughtful comments. We respectfully address the concerns below.
>
> 1. Operational implications
>
> We appreciate the feedback on providing more operational and executable implications. In response, we have **provided an example of a geometric property to use** and **analyzed its correlation with robustness** at the end of **Section 4.2**. We show that the average difference between positive vs. negative embeddings is well-correlated with robustness to noise and is a measure that would be easy to compute in practice.
>
> 2. Model-dependence
>
> Thank you for this comment, and we have
> 1. Clarified that **separation is model-dependent** and **separability should be evaluated at the beginning of training** in the **implications section**.
> 2. Discussed how **low separation can occur despite clear preferences due to the model's representations** in the **implications section**.
> 3. Performed the **experiment across multiple models** in **Section 4.2** and found that instruction-tuned models have the best performance/robustness while smaller models have the weakest robustness. However, we could not include a sentence embedding model such as BERT, as we cannot perform DPO on these models.

---

### Author Response · Authors · 2025-11-22

We would like to thank the reviewers and chairs for their time and effort in providing feedback and suggestions on our work.

This work examines the **impact of human errors and inconsistencies** on preference learning from a theoretical perspective. We derive robustness bounds under models of **real-world sources of noise** such as mislabeling and uncertainty. Our results show how the structure of the embedding space influences the rate at which performance degrades with noise, offering insights for selecting preference learning algorithms.

We are glad that reviewers recognize our paper to have **rigorous theoretical analysis** with **empirical alignment**, providing **practically effective** results for a topic **central to current alignment practices**.

### Key Additions and Changes in the manuscript

In response to reviewer feedback, we have:

1. Expanded experiments to compare **robustness across various models** such as instruction-tuned models and smaller models
2. Clarified that separability is also model-dependent
3. Provided further guidance on **how to evaluate separability** as well as an example metric that can be used
4. Expanded discussion of theoretical results to include expectations for how they will **generalize under different assumptions**

We believe that with these changes, key concerns have been addressed and the rigor and clarity of our work have been strengthened. Thank you again for your efforts, and we hope these remarks help with your discussion.

---

### Decision · Action_Editor_38MY · 2026-01-05

**Recommendation:** Accept as is

**Additional Comments:**

This paper provides a theoretical and empirical analysis of preference optimization under noisy feedback. Specifically, the authors provide generalization bounds for a unified Generalized Preference Optimization (GPO) formulation, where the bounds depend on dataset size, embedding concentration/separation, and noise level. The authors validated the theoretical results using synthetic data and Anthropic
Evaluations datasets. All reviewers agree that the paper is technically sound and provides valuable insights into why certain models are better at handling noisy human feedback than others. While all reviewers agree to accept the paper from a merit perspective, there is a concern regarding anonymity: reviewer noticed an anonymity violation in the revised manuscript where the author names are revealed.  AE believes the technical evaluation will reach the same consensus given the positive review on initial anonymized manuscript. Given that all reviewers (including the reviewer who noticed the violation) agree that the technical concerns have been fully addressed and the paper is of high quality, this does not preclude acceptance. After consulting with the Editors-in-Chief, AE recommends acceptance of the paper.

**Audience:**

Yes

**Audience Explanation:**

The theoretical framework and findings would be interesting to the audience who work on preference learning and LLM alignment.

**Claims And Evidence:**

Yes

**Claims Explanation:**

The claims are supported by the theoretical analysis of the impact of noisy feedback. The authors also validated the analysis with empirical results.